# Neurosymbolic Language Reasoning as Satisfiability Modulo Theory

## Abstract

Natural language understanding requires interleaving textual and logical reasoning, yet large language models often fail to perform such reasoning reliably. Existing neurosymbolic systems combine LLMs with solvers but remain limited to fully formalizable tasks such as math or program synthesis, leaving natural documents with only partial logical structure unaddressed. We introduce Logitext, a neurosymbolic language that represents documents as natural language text constraints (NLTCs), making partial logical structure explicit. We develop an algorithm that integrates LLM-based constraint evaluation with satisfiability modulo theory (SMT) solving, enabling joint textual–logical reasoning. Experiments on a new content moderation benchmark, together with LegalBench and SuperNatural Instructions, show that Logitext improves both accuracy and coverage. This work is the first that treats LLM-based reasoning as an SMT theory, extending neurosymbolic methods beyond fully formalizable domains.

## 1 Introduction

Large language models (LLMs) remain unreliable at logical reasoning in natural language, often producing inconsistent or incomplete results despite recent progress [Sakai et al. (2025); Lin et al. (2025)]. Logical solvers provide reliable guarantees but are confined to fully formalizable domains such as math and program synthesis. Existing neurosymbolic systems combine LLMs with logical solvers to achieve strong results in these domains [Olausson et al. (2023); Ye et al. (2023); Wen et al. (2025)], but they face two key limitations. First, they remain restricted to fully formalizable settings and thus cannot naturally handle documents that mix textual and logical structure. Second, they typically adopt a staged architecture in which the LLM formalizes the problem once and the logical solver executes it. This design precludes the iterative interleaving of textual and logical reasoning required for many natural language tasks.

Real-world documents highlight this gap. Policies specify conditions on user posts, instructions impose formatting rules, and statutes constrain legal interpretations. These constraints are seldom fully formalizable, yet they combine naturally with logical operators and interact with calculations. To capture such cases, we introduce **Logitext**, a neurosymbolic language that expresses constraints directly in text. At the core of Logitext are *natural language text constraints (NLTCs)*, a representation that makes partial logical structure explicit and allows textual and symbolic constraints to work together.

*Example.* Consider a policy stating that "a post must be removed if it is both hateful and an immediate threat." The textual notions of "hateful" and "immediate threat" cannot be fully formalized in logic, but they can be represented as NLTCs. Logitext links these textual constraints with a logical conjunction, ensuring that the decision depends on both the logical structure and the outcome of the textual judgments.

We realize this idea by extending satisfiability modulo theory (SMT) with a new theory for textual constraints. Modern SMT logical solvers assign values to variables step by step and propagate the consequences across theories such as strings, floats, and sets [Davis et al. (1962), Marques-Silva & Sakallah (1999), de Moura & Bjørner (2008), Barrett et al. (2010), Zheng et al. (2017), Rümmer & Wahl (2010)]. With NLTCs, this propagation requires

Messages $^M$ containing disruptive behavior are those $^{C_1}$ addressed at a group (not just an individual), **where** $^{C_2}$ the group targeted by the message is defined by ethnicity, gender, color, nationality, sexual orientation, race, or physical disability, **and** the message matches **at least one of** the following sub-rules:

- Bias: $^{C_3}$ Message contains stereotyping, insensitive remarks, fear of difference, non-inclusive language, microaggressions, justifying biases by seeking out like-minded people, accepting negative or misinformation/screening out positive information.

- Violence: $^{C_4}$ Message is related to murder, rape, assault, arson, terrorism, vandalism, desecration, or threats.

- Genocide: $^{C_5}$ Message is related to the act or intent to deliberately and systematically annihilate an entire people.

Based on the above, a message is an immediate threat if $^{C_6}$ it expresses a violent or genocidal intention and the context is enough to suggest that the safety and/or life of an individual or group of people is at risk.

(a)

Check if the following message $M$ contains disruptive behavior [or an immediate threat] as per the above policy: …

(b)

Create a sample message $M$ that contains disruptive behavior as per the above policy. Ensure that the message does not involve violence or genocide.

(c)

Create sample messages $M$ that each contain disruptive behavior according to the policy. Ensure that the messages do not involve violence or genocide. Create a sample for every valid combination of policy criteria.

(d)

Figure 1: Example: A content moderation policy (a) illustrates a fine-grained mix of logical and textual constraints. Combined with (b), it yields a prompt that requires compositional logical reasoning, and with (c-d), combinatorial reasoning.

solving textual constraints iteratively so that assignments remain consistent with Boolean conditions. We develop a theory that performs this process efficiently and integrate it into existing SMT solvers, thereby positioning LLM-based reasoning as an SMT-solver-compatible theory.

**Contributions.** This paper formalizes LLM-based reasoning as an SMT theory and introduces the first framework to support SMT-solver-compatible reasoning with partial logical structure:

- **Concept:** We show the necessity of interleaving textual and logical reasoning and characterize the limitations of staged approaches (§2).
- **Language:** We introduce Logitext, a neurosymbolic language that enriches documents with logic and represents them as NLTCs interfacing with SMT solvers (§3.1, §3.2).
- **NLTC Solver:** We present an algorithm that solves NLTCs and extends the SMT framework with this capability (§3.3).
- **Evaluation:** We show that Logitext outperforms staged baselines on a new content moderation benchmark and improves accuracy and coverage on LegalBench and SuperNatural Instructions (§4).

## 2 Interleaved text/logic reasoning in text understanding

We illustrate the need for interleaved textual and logical reasoning using a content moderation policy.

### 2.1 Compositional vs combinatorial reasoning in natural text

Figure 1 shows a common LLM use case. A policy document (Figure 1a) defines notions such as "disruptive behavior" and "immediate threat." When paired with a task description

(Figures 1b–1d) and an input message, the document becomes an LLM prompt whose **reliable reasoning** is the objective.

The document expresses intent through both textual and logical relations. Shaded text *within* each clause $C_i$ specifies its *meaning* relative to the input message $M$ and possibly other clauses. For example, given $M =$ "Americans love ice cream," clause $C_1$ ("addressed at a group") and $C_2$ ("targeted by nationality") both evaluate to True. Underlined text *between* clauses then constrains these meanings logically. Formally, whether a message is disruptive ($d$) depends on:

$$d = [[C_1]] \wedge [[C_2]] \wedge ([[C_3]] \vee [[C_4]] \vee [[C_5]]) \tag{1}$$

**Compositional reasoning.** The classification task in Figure 1b asks whether a message is disruptive ($d$) or an immediate threat ($t$). Solving for $d$ requires one pass of textual reasoning to evaluate each $[[C_i]]$, followed by logical evaluation of the formula. At first glance, $t$ appears to need only textual reasoning ($[[C_6]]$). However, $C_6$ checks whether the message "expresses violent or genocidal intention," which depends on prior results $[[C_4]]$ and $[[C_5]]$. Thus, $t$ requires information from a logical disjunction $[[C_4]] \vee [[C_5]]$, showing the benefit of interleaving logical with textual reasoning.

**Combinatorial reasoning.** The constrained generation task in Figure 1c reverses the classification problem: instead of labeling a given message, the goal is to generate a message $M$ that satisfies both a partial assignment of clause values and the policy as a whole. This requires two steps. First, a logical solver proposes candidate assignments for the relevant clauses (e.g., $C_1 \ldots C_5$) that are consistent with the partial assignment and with the logical definition of $d$. Second, a generator synthesizes a message $M$ whose text realizes those clause assignments. If the synthesis step fails to produce a valid message, the process must repeat with a new candidate assignment. The high-coverage generation task in Figure 1d is an even harder variant: it requires generating messages that realize many or all satisfying assignments, not just one. Such tasks inherently demand iterative cooperation between logical solving and textual synthesis.

## 2.2 Logical reasoning gaps

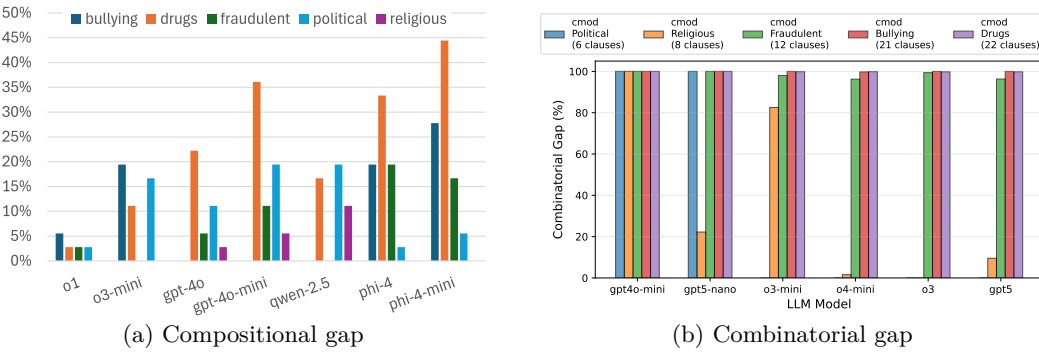

(a) Compositional gap          (b) Combinatorial gap

Figure 2: Gaps in logical reasoning (See App A.1.) on content moderation across LLMs.

How well do LLMs perform the logical reasoning needed for these tasks? To evaluate this, we define **reasoning gaps**: the difference between (1) performance when an LLM handles textual reasoning and a logical solver executes logical reasoning, versus (2) performance when the LLM alone handles the entire prompt. We denote the compositional gap by $\Delta$ and the combinatorial gap by $\Delta'$. Our dataset contains five content safety policies with 6–21 clauses.

Figure 2a shows $\Delta$ across models, measured as classification accuracy. Smaller models exhibit large compositional gaps, while stronger models narrow them. Within each model, the size of the gap increases with task complexity, reflecting the number of interacting clauses.

Figure 2b shows $\Delta'$ for combinatorial reasoning, measured as the number of distinct correct assignments generated. Here the gaps are dramatic: even state-of-the-art models such as

```
```(M:str, d, t, C1, C2, C3, C4, C5, C6, C7)```
```

Messages containing disruptive behavior are those {{let C1 = [[addressed at a group (excludes messages targeted at just an individual)]] where "messages" is M}} and {{let C2 = [[the group targeted by the message is defined by {{forsome [[ethnicity]], [[gender]], [[color]], [[nationality]], [[sexual orientation]], [[race]], or [[physical disability]] }}]] where "message" is M}}, and the message matches at least one of the following sub-rules:

Bias: {{let C3 = [[Message contains stereotyping, insensitive remarks, fear of difference, non-inclusive language, microaggressions, justifying biases by seeking out like-minded people, accepting negative or misinformation/screening out positive information.]] where "message" is M}}

Violence: {{let C4 = [[Message is related to murder, rape, assault, arson, terrorism, vandalism, desecration, or threats.]] where "message" is M}}

Genocide: {{let C5 = [[Message is related to the act or intent to deliberately and systematically annihilate an entire people.]] where "message" is M}}

```
C8 = C4 or C5
```

Based on the above, a message is an immediate threat if {{let C6 = [[it expresses a violent or genocidal intention]] where "it expresses a violent or genocidal intention" is C8}} and {{let C7 = [[the context is enough to suggest that the safety and/or life of an individual or group of people is at risk.]] where "the context" is M }}

```
d = C1 and C2 and (C3 or C4 or C5)
t = C6 and C7
```

Figure 3: Content moderation policy example implemented as Logitext Document

GPT-5 fail to recover over 99% of satisfying assignments that an SMT solver (Z3) can enumerate, and GPT-4o-mini fails completely across all tasks. Unlike compositional gaps, which shrinks with model scale, combinatorial gaps remain severe even for frontier models.

In summary, although improvements in models gradually address compositional gaps, combinatorial gaps (which affect the solve/synthesize loop of language reasoning) are still significant. Logitext is designed to bridge these gaps by helping specify textual vs logical intent of natural documents precisely, and finely interleave LLM decoding and logic solving to support combinatorially efficient and semantically faithful interpretation of the intent.

## 3 LOGITEXT: LANGUAGE, REPRESENTATION AND SOLVER

Given a conventional textual prompt as in Figure 1, we convert it to Logitext program format by annotating it (Section 3.1). The Logitext program is *parsed* into set of hybrid *Natural Language Text Constraints* (NLTCs) (Section 3.2). Section 3.3 presents an LLM-based solver NLSolver for NLTCs and pair it with a logical solver to produce the final task response.

### 3.1 THE LOGITEXT LANGUAGE

Logitext extends conventional text prompts into *hybrid text/logic documents*, enabling natural language clauses to interact directly with formal constraints. It supports *partial formalization*: only those parts of a document that benefit from logical structure are annotated, while the rest remain textual. This selective annotation allows reasoning to interleave between textual interpretation and logical propagation, as motivated in Section 2.1.

A Logitext document (Figure 3) enriches a textual policy (Figure 1a) with four constructs (see Appendix A.2 for the full syntax):

- **Variable declarations** (e.g., `(M:str, d, t, …)`) define the symbols that participate in logical constraints. Variables may be Boolean or string; string variables must be typed explicitly (e.g., `M:str` for an input message).

- **Textual let bindings** of the form `{{let <var_0> = [[<clause>]] where <subclause_1> is <var_1> ... and <subclause_n> is <var_n>}}` (a) binds a textual

clause (i.e., a sentence fragment) to a logical variable (for example, the clause "addressed at a group ...just an individual" is named `C1`), and (b) associates sub-clauses within the clause (e.g., "messages") with external variables, e.g. `M`. Intuitively, `<clause>` represent a constraint between the variables `<var_i>`.

- **Logical constraint blocks** (delimited by ` ``` `) specify logical relations (e.g., `t = C6 and C7`) among variables, using `pyz3` notation [z3p; de Moura & Bjørner (2008)].

- **Convenience constructs** such as `forall` and `forsome` compactly handle textual lists, internally expanded into disjunctions or conjunctions over let-bindings.

Such Logitext documents are "executed" using a `check()` function as in constraint solving. Given a partial assignment `p` of variables in a document `d`, `check(d, p, cover)` searches for a satisfying assignment that respects both the logical and textual constraints:

```
check(d:LogitextDocument, p:Dict[str, bool|str], cover:Option[bool])
-> Dict[str, bool|str] | unsat | timeout
```

If a solution exists, `check()` returns a full assignment as a mapping from variable names to values. Otherwise it reports unsatisfiability or timeout. With the optional flag `cover`, `check()` enumerates multiple satisfying assignments. This mechanism generalizes the familiar `complete()` execution of text prompts to a richer constraint-satisfaction setting.

The expressiveness of Logitext unifies diverse language understanding tasks under a single interface. The three tasks of Figure 1(b)–(d) are expressed uniformly as constraint checking: (i) **classification** (`lt.check(d, {'M': M})['d']`), (ii) **partially constrained instance generation** (`lt.check(d, {'C4': False, 'C5': False})['M']`), and (iii) **coverage generation** (`[g['M'] for g in lt.check(d, {'C4': False, 'C5': False}, cover=True)]`). In contrast to raw prompting, Logitext makes explicit the logical structure of documents, enabling solver-style propagation to cooperate with LLM-based textual reasoning.

## 3.2 Natural language text constraints

The constructs in Section 3.1 define how Logitext documents combine textual clauses with logical constraints. To reason with such documents, we require a representation that treats textual clauses as first-class objects alongside logical formulas. We introduce *natural language text constraints (NLTCs)*, which bind clauses to variables, record references to external context, and allow seamless interaction with solvers.

Recall from the previous section that, in addition to a variable declaration section, an unparsed Logitext document $d$ consists of alternating code blocks and text blocks (Figure 3). Each code block is a sequence of logical strings $k$, e.g., `C8 = C4 or C5`. Each text block contains a sequence of let-binding text strings $L$ of the form:

$$\textbf{let } v = [[c]] \textbf{ where } u_1 \textbf{ is } p_1 \ldots u_n \textbf{ is } p_n.$$

Here $v$ is a boolean variable to which $c$ is bound, while the $u_i$ are strings (typically substrings of $c$) associated with variables $p_i$ defined elsewhere.

To process a document $d$, we parse it into an abstract representation $D$ in three steps:

1. **Variable collection.** Identify boolean variables $vs_D = v_1, \ldots, v_n$ and string variables $us_D = u_1, \ldots, u_{n'}$. These variables include those declared explicitly and those introduced in let bindings as above.

2. **Logical constraint parsing.** Convert each logical string $k$ from a logical text string into a solver-ready formula $\phi$ using Z3's parser.

3. **Textual constraint parsing.** Translate each let-binding $L$ into an NLTC $\nu = (v, c, \{u_1 : p_1, \ldots, u_n : p_n\}, d)$: each NLTC binds $v$ to the clause $c$, records its dependencies, and points to the full document $d$ so $c$ can be interpreted in context.

After parsing, the abstract document is $D = (vs_D, us_D, \phi_D, \nu_D)$, where $\phi_D = \phi_1, \ldots, \phi_m$ are logical constraints and $\nu_D = \nu_1, \ldots, \nu_n$ are NLTCs. Reasoning proceeds with respect to a partial assignment $\pi_D : us_D \cup vs_D \to \text{bool|str}$, which specifies known variable values and lets the solver–LLM loop infer the rest, as discussed in the next section.

## 3.3 Solving natural language text constraints

| **Algorithm 1** check($D = (\nu, \phi, vs, us), \pi_D$) | NLSolver(u, $\nu$, $\pi$) |
|---|---|
| 1: **while** true **do** | 1: $u^* \leftarrow$ LLMPropose($\nu$, $\pi$, {}, None)     ▷ Propose |
| 2:    $\pi_Z \leftarrow Z3(\phi, vs, \pi_D)$ ▷ Propose bool. assignment | 2: **for** $t = 1$ to $T$ **do** |
| 3:    **return** UNSAT **if not** $\pi_Z$ | 3:    **for** $\nu_k \in \nu$ **do with** sat $\leftarrow$ true; $\bar{\pi} \leftarrow$ {} |
| 4:    **for** unbound $u \in us$ **do with** sat $\leftarrow$ true; $\pi_s \leftarrow$ {} | 4:        ▷ Verify proposal; record unsatisfied clause |
| 5:        $u^* \leftarrow$ NLSolver(u, $\nu[u]$, $\pi_D \cup \pi_s \cup \pi_Z$) | 5:        **if** LLMVerify($\nu_k$, $\pi \cup \{u = u^*\}$) $\neq \pi[\nu_k]$ **then** |
| 6:        **if** !$u^*$ **then** $Z3$.block($\pi_Z$) ; sat $\leftarrow \varnothing$ ; **break** | 6:            sat $\leftarrow$ false ; $\bar{\pi} \leftarrow \bar{\pi} \cup \{\nu_k\}$ |
| 7:        $\pi_s \leftarrow \pi_s \cup \{u = u^*\}$ | 7:        **If** sat **then return** $u^*$ |
| 8:    **if** !sat **then continue** | 8:    $u^* \leftarrow$ LLMPropose($\nu$, $\pi$, $\bar{\pi}$, $u^*$)      ▷ Refine |
| 9:    **return** $\pi_s \cup \pi_D \cup \pi_Z$ | 9: **return** None |

| (a) Outer logical solver loop | (b) Inner text solver loop |
|---|---|

Figure 4: Core Logitext constraint solving algorithms

Figure 4 shows how Logitext solves hybrid systems of natural language text constraints (NLTCs, $\nu$) and logical constraints ($\phi$), given shared Boolean variables $vs$, text-string variables $us$, and a partial assignment $\pi_D$ of Booleans and strings. The goal is to extend $\pi_D$ into a complete satisfying assignment $\pi_D'$, using an LLM-based solver as an extension to the core logical (aka SMT) solver.

The overall strategy is simple. A logical solver (we use $Z3$ de Moura & Bjørner (2008)) produces candidate Boolean assignments that satisfy Boolean constraints (Fig 4a), and our LLM-based solver (called NLSolver) then attempts to produce assignments to string variables that satisfy text constraints while maintaining the Boolean assignments (Fig 4a).

We begin with the outer logical solver loop of Fig 4a. We use $Z3$ to generate candidate assignments $\pi_Z$ of variables $vs$ consistent with $\phi$ and $\pi_D$ (Line 2). We now loop sequentially over unbound string variables $u$ (Lines 4-9), trying to find satisfying assignments for each, compatible with all assignments so far. Each $u$ is passed to text-constraint solver NLSolver, which attempts to generate a text string value $u^*$ for each unbound string variable $u$ (Line 5), given the constraints $\nu[u] \subseteq \nu$ that read or write $u$, and the assignments so far ($\pi_s \cup \pi_D \cup \pi_Z$). If NLSolver fails to find a $u^*$, we block $Z3$ from regenerating candidate assignment $\pi_Z$, break out of the loop over $us$ (Line 6) and continue generating more candidate Boolean assignments (Lines 8, 2). If all string variables $u$ are assigned, we declare success and return all assignments accumulated (Lines 7, 9). If no candidates remain, we declare unsatisfiabilty (Line 3).

NLSolver (Fig. 4b) is given a variable $u$, a set $\nu$ of NLTCs that read $u$, and a partial assignment $\pi$. Its job is to produce a textual string $u^*$ for $u$ that satisfies constraints $\nu$ given partial assignment $\pi$. It does so through a propose-verify-refine loop. It starts by prompting an LLM, via the LLMPropose() call (see Appendix 3), to propose a candidate $u^*$ that satisfies $\nu$ and $\pi$ (Line 1). It then uses $T$ rounds (Line 2) to refine this solution to one that satisfies $\nu$. In each round, for every NLTC $\nu_k \in \nu$, it calls into an LLM via LLMVerify() (Appendix **??**) to infer the truth value for the variable bound by $\nu_k$, given $u = u^*$ and existing assignment $\pi$, and compares this truth value to that required by the partial assignment $\pi$ (Line 5). If all truth values are compatible, it returns $u^*$ as a satisfying assignment (Line 7). Otherwise, it uses LLMPropose() to refine $u^*$ (Line 8). The refinement is guided by an additional "needs-to-change" set $\bar{\pi}$, which lists the constraints that $u^*$ currently violates. After $T$ rounds of not finding $u^*$, we declare failure (Line 9).

The above describes the essence of how Logitext solves NLTCs. In practice, we include two further techniques that have modest impact. First, note that LLMPropose() may produce a piece of text that does not match the current partial assignment $\pi$, and is therefore rejected by LLMVerify(). However, LLMPropose() itself may be called many thousands of times for various candidate assignments in the check() algorithm, Line5. Given that calls to LLMPropose() are relatively expensive since it calls out to LLMs, we cache results from these calls and consider the for use on future calls to NLSolver. Second, when we propose a refinement of textual value $u^*$, it helps not only to have the "needs-to-change" list mentioned above, but also a history of the previous refinements proposed on $u^*$ and their outcome from LLMVerify. These two techniques are described further in the appendix, and the (modest but noticeable) impact of caching is analyzed in the evaluation section (Figure 6).

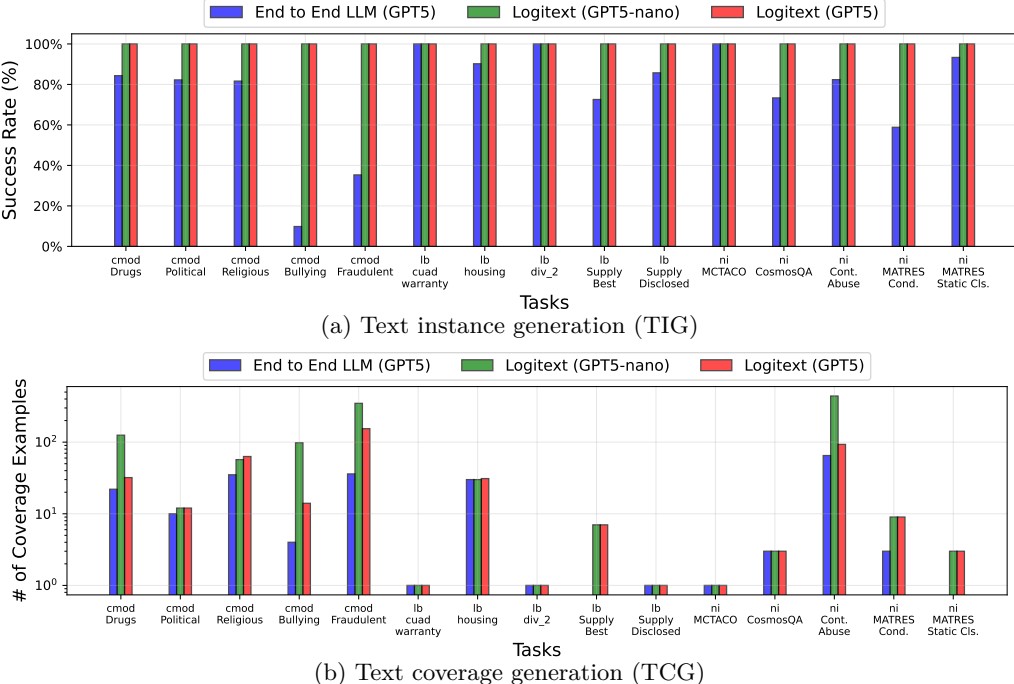

(a) Text instance generation (TIG)

(b) Text coverage generation (TCG)

Figure 5: End-to-End performance comparison per task.

# 4 EVALUATION

## 4.1 BENCHMARKS AND SETUP

**Content Moderation (CMOD).** A new benchmark of five multi-page moderation policies (2–5 pages, 6–22 annotated clauses) covering drugs (22 clauses), politics (8), religion (6), bullying (21), and fraud (12) (see Appendix A.9 for an example). These tasks are designed to reflect realistic compliance settings where policy documents constrain user-generated content.

**Legal Benchmark (LegalBench, LB).** We select five tasks from LegalBench [Guha et al. (2023)], an extensive benchmark for reasoning over statutory and regulatory text. The tasks cover domains such as diversity jurisdiction, housing and warranty law, and supply-chain transparency. Each task is 20–50 lines with 2–4 annotated clauses. This benchmark captures challenges in legal text where precise logical structure interacts with natural language.

**Natural Instructions (NI).** We select five tasks from SuperNatural Instructions [Wang et al. (2022)], focusing on problems with implicit logical constraints such as detecting grammatical inconsistencies, reasoning about hypothetical actions, and identifying abusive content. Each task is 3–10 lines with 2–5 annotated clauses. This benchmark tests generalization to diverse instruction-following tasks beyond policy or law.

Together these benchmarks yield 15 tasks with 10+ instances each, spanning policy, legal, and open-domain instructions. All tasks require mapping text inputs to structured outputs (classifications or constrained generations). We evaluate Logitext on three settings aligned with Fig. 1: (a) **Task execution (TE)** — measuring classification accuracy on task instances, (b) **Text instance generation (TIG)** — testing the ability to generate valid inputs under partial constraints, and (c) **Text coverage generation (TCG)** — enumerating as many valid inputs as possible.

## 4.2 RESULTS

**Text instance generation (TIG).** Figure 5a compares Logitext with direct prompting. With both GPT5 and GPT5-nano as base models, Logitext generates valid assignments reliably via `check()`. The results indicate that (i) Logitext attains near-saturation perfor-

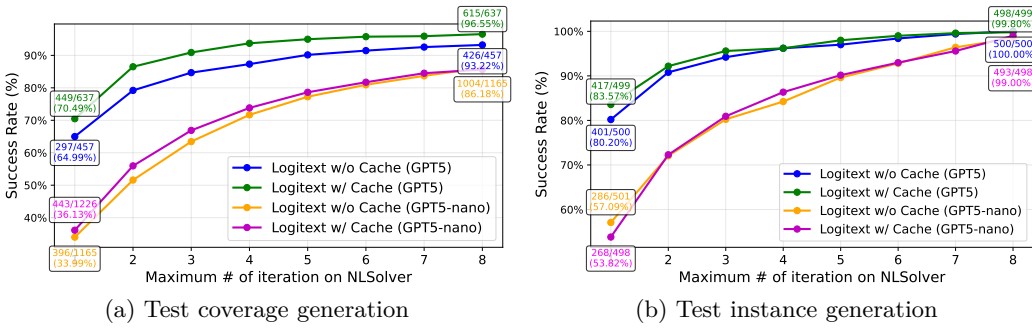

(a) Test coverage generation      (b) Test instance generation

Figure 6: NLSolver success rate vs num. iterations.

mance even with the weaker GPT5-nano, while direct prompting to GPT5 shows noticeable degradation; and (ii) degradation is most evident on complex CMOD policies, although performance on Drugs is relatively stronger than other cases.

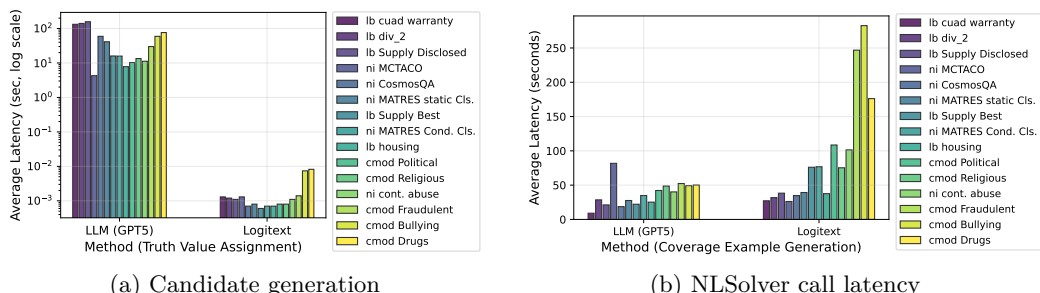

(a) Candidate generation      (b) NLSolver call latency

Figure 7: Component-wise latency on TCG (tasks sorted by the #clauses).

**Text coverage generation (TCG).** Figure 5b (log scale) reports coverage under a fixed time budget of 3000s. Baseline GPT is allowed up to 5 iterations for candidate generation and 5 additional iterations for finding satisfying assignments. Logitext achieves broader coverage, particularly with GPT5-nano. Figure 7 provides an explanation: candidate generation is significantly faster with Logitext since it uses a solver rather than repeated LLM calls (Fig. 7a). The advantage is reduced in the solving phase (Fig. 7b), where NLSolver calls dominate, but this bottleneck is smaller for faster base models such as GPT5-nano. We also report the LLM call statistics of NLSolver in Figure 11 of Appendix.

**NLSolver iterative refinement.** Figure 6 shows that NLSolver improves success rates as the number of iterations increases. Caching can be beneficial in some settings (e.g., coverage generation with GPT5), though its overall effect across tasks is limited.

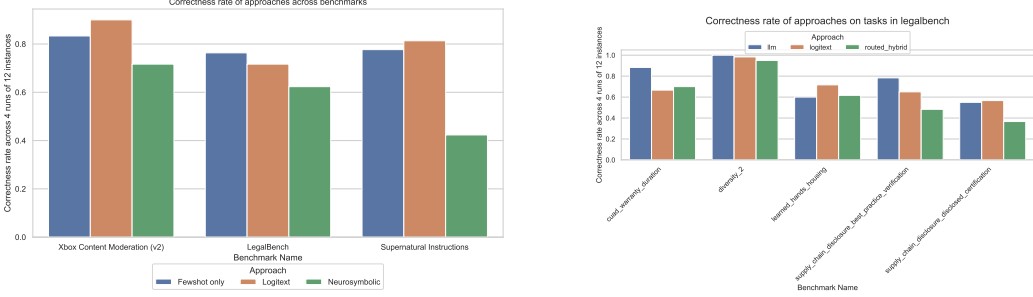

Figure 8: Aggregated Task Execution (TE) Result (left), Legalbench in detail (right).

**Task Execution (TE).** Figure 8(left) presents accuracy on TE tasks using GPT-4o. In addition to few-shot prompting, we include a neurosymbolic prompt that generates and executes code. Logitext performs better on CMOD and NI benchmarks but underperforms on LegalBench. Figure 8(right) highlights two main sources of error: (i) in some cases, clause-level outputs $[[C_i]]$ were incorrectly predicted by the LLM, but the LLM-only approach produced correct answers using holistic reasoning, revealing a robustness limitation for Logitext; (ii) in other cases, lists such as "x, y, and z" were intended as examples rather than conjuncts, but were annotated as the latter. These issues point to the need for clause-level error correction and more careful handling of list annotations in future work.

Overall, the experiments show that Logitext improves performance on both constrained generation and classification tasks across multiple benchmarks, while highlighting remaining challenges in clause-level robustness and annotation handling.

## 5 Related work

**Prompt-based reasoning.** Prompting strategies such as Chain-of-Thought (CoT) [Wei et al. (2022)] and Tree-of-Thought (ToT) [Yao et al. (2023)] elicit multi-step reasoning by decomposing queries into textual steps. Chain-of-Logic [Servantez et al. (2024)] separates logical reasoning from answer prediction, aiming to improve consistency in step-wise deduction. While these approaches enhance local coherence or allow limited backtracking, they lack mechanisms to bridge deeper compositional and combinatorial reasoning gaps and cannot ensure global logical consistency across clauses. Once an error propagates, there is no principled way to validate against constraints. Our framework differs by explicitly defining these reasoning gaps and incorporating symbolic validation into the reasoning process itself.

**Systematic generalization and constraint solving.** Our challenges relate closely to systematic generalization tasks [Lake & Baroni (2018), Keysers et al. (2020), Kim & Linzen (2020)], which demonstrate that sequence models fail when compositional rules must be recombined in novel ways. Similar issues arise in program synthesis and constraint satisfaction tasks, where LLMs can propose candidate programs or assignments (e.g., Codex for SAT/SMT) but collapse under combinatorial growth in the search space. These methods provide no principled mechanism to enforce or recover from violated constraints. We formalize these compositional and combinatorial reasoning gaps as structural limitations of LLM inference and show how solver integration can systematically mitigate them in natural language contexts.

**Reasoning models.** Reasoning models such as OpenAI o3/o4-mini and DeepSeek-R1 [Guo et al. (2025)] have shown improved performance on benchmarks for logical reasoning and robust instruction following. RL allows limited correction through feedback [Kalyanpur et al. (2024)] or exploration [Xie et al. (2025)]. However, they depend heavily on reward shaping or sample filtering, lack a formal representational layer for partial logical structures. As a result, they cannot enforce symbolic constraints or recover from violated ones during inference. Our framework complements these advances by introducing a neurosymbolic language that supports constraint-aware reasoning within an SMT framework.

**Neuro-symbolic reasoning.** Recent systems such as LINQ [Olausson et al. (2023)], CLOVER [Ryu et al. (2025)], and ZebraLogic [Lin et al. (2025)] connect LLMs with symbolic solvers by translating natural language into executable logic programs. These approaches achieve strong guarantees when tasks are fully formalizable, but their reliance on complete logical structure restricts applicability to natural documents like policies or legal texts, where only fragments of logic are explicit. ZebraLogic also provided an initial study of the *combinatorial gap*, but its scope was limited to well-defined mathematical domains. In contrast, our work addresses this challenge in natural language contexts that inherently contain uncertainty and partial structure. We introduce natural language text constraints (NLTCs), enabling partial formalization and iterative solver-guided reasoning that better reflects the complexity of real-world documents.

## 6 Conclusion

In this work we introduced **Logitext**, a neurosymbolic framework that treats LLM reasoning as an SMT theory through natural language text constraints. We first motivated the need for such a framework by showing that even frontier LLMs continue to exhibit two reasoning gaps: compositional gaps that narrow with scale but persist, and combinatorial gaps that remain severe. By interleaving solver propagation with LLM decoding, Logitext provides a principled way to reduce these gaps. More broadly, our results suggest a path toward positioning LLMs as solver-compatible theories, opening opportunities for scalable, reliable, and trustworthy natural language reasoning.

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

## A    APPENDIX

Additional information follows.

### A.1    DEFINITIONS OF GAPS

Each policy $p$ is annotated with clauses $C_i$ and associated with a formula $\phi$ as in Fig 1 and Eqn1, and comes with 10-20 test messages $M_j$ each with ground truth $H_j$.

**Definition 1** (**Compositional gap $\Delta_{mp}$ of LLM $m$ on prompt $p$**). For each $M_j$, prompt $m$ with $p$ for (i) the meanings $b_{ij}$ of clauses $C_i$, and (ii) whole-prompt result $h_j$. This results in overall accuracy $a = \text{mean}_j \delta(h_j, H_j)$, where $\delta(x, x') = 1$ if $x = x'$, else 0. Now, use a logical solver to evaluate $h_j^* = \phi(b_{ij})$, giving corresponding accuracy $a^*$. Then compositional gap $\Delta_{mp} = a - a^*$

**Definition 2** (**Combinatorial gap** $\Delta'_{mp}$ **of model LLM** $m$ **on prompt** $p$). Prompt $m$ with $p$ to produce (i) all policy inputs $M_j$, and (ii) complete assignments $b_{ij}$ for clauses $C_i$ such that the policy is true. Use a logical solver to filter out $b_{ij}$ that do not satisfy $\phi$, and also to generate independently its satisfying assignments $b^*_{ij}$. Let $n_j$ (resp. $n^*_j$) be number of such assignments . The combinatorial gap is the relative discrepancy between these numbers: $\Delta'_{mp} = \text{mean}_j(n^*_j - n_j)/n^*_j$,

## A.2 Syntax for Logitext documents

$$\textbf{doc } d \leftarrow [b \mid c]^+$$
$$\textbf{text block } b \leftarrow [s \mid t]^+$$
$$\textbf{code block } c \leftarrow \text{```}[(v1, \ldots, vn)]\langle\text{code}\rangle\text{```}$$
$$\textbf{text } s \leftarrow \langle\text{strings without \{\{ or \}\}}\rangle$$
$$\textbf{term } t \leftarrow l \mid q$$
$$\textbf{let } l \leftarrow \{\{ \text{ let } v = [[b]] \text{ where } r_1 \text{ is } v_1 \text{ and } \ldots \text{ and } r_n \text{ is } v_n \}\}$$
$$\textbf{quantifier } q \leftarrow \{\{\text{forall } [s \mid [[b]]]^+ \}\} \mid \{\{\text{forsome } [s \mid [[b]]]^+ \}\}$$
$$\textbf{typed variable } v \leftarrow \langle\text{variable name}\rangle[: \text{str}]$$
$$\textbf{quoted str. } r \leftarrow \texttt{"..."}$$

## A.3 Expanded dataset

| Benchmark | Task | Original Submission | | Resubmission | | Evaluated |
|---|---|---|---|---|---|---|
| | | #Inst | #Runs | #Inst | #Runs | #Inst |
| **cmod** | | | | | | |
| Bullying_2.0 | | 12 | 5 | 100 | 5 | 0 |
| Drugs_&_Alcohol_2.0 | | 12 | 5 | 100 | 5 | 0 |
| Fraudulent_v2.0v | | 12 | 5 | 100 | 5 | 0 |
| Political_v2.0v | | 12 | 5 | 100 | 5 | 0 |
| Religious_v2.0v | | 12 | 5 | 100 | 5 | 0 |
| **legalbench** | | | | | | |
| cuad_warranty_duration | | 12 | 5 | 100 | 5 | 100 |
| diversity_2 | | 12 | 5 | 100 | 5 | 100 |
| learned_hands_housing | | 12 | 5 | 100 | 5 | 100 |
| supply_chain_disclosure_best_practice_verification | | 12 | 5 | 100 | 5 | 100 |
| supply_chain_disclosure_disclosed_certification | | 12 | 5 | 100 | 5 | 100 |
| **natural_instructions** | | | | | | |
| task021_mctaco_grammatical_logical | | 12 | 5 | 100 | 5 | 50 |
| task022_cosmosqa_passage_inappropriate_binary | | 12 | 5 | 100 | 5 | 50 |
| task108_contextualabusedetection_classification | | 12 | 5 | 100 | 5 | 50 |
| task457_matres_conditional_classification | | 12 | 5 | 100 | 5 | 50 |
| task459_matres_static_classification | | 12 | 5 | 100 | 5 | 50 |

Table 1: Comparison of dataset instance counts and runs in the original submission vs. resubmission.

We have expanded the dataset evaluated to 100 samples per task from 12 samples per task as shown in Table 1. We are in the process of running evaluations on the data. So far, we have completed full re-evaluation on 100 samples on 5 tasks (from Legalbench), partial re-evaluation on 50 samples on (from Supernatural Language Instructions (SNLI)), and have not yet started re-evaluation on our most favorable dataset CMOD. For Legalbench and SNLI, each task had sufficient samples that we were able to simply incorporate more samples from the existing dataset. For CMOD, we had to generate samples analogous to production moderation data, a task that requires some care.

We focused on re-running the Task Execution (TE) experiments (Figure 8), both because these are the least favorable to us, and because the combinatorial gap experiments take much longer to run. Of course, we will complete running all experiments on all datasets over the next few days.

As Tables 2 and 3 show, the expanded results don't change the highest level message on the Task Execution task qualitatively: Logitext does provide a noticeable boost on many tasks, and prevails in 7 of 10 tasks, but the baseline model does do better in some cases. Perhaps interestingly, Logitext now does relatively better on Legalbench, prevailing in 4/5 tasks, and slightly worse on SNLI (3/5). Once again, when Logitext fails, the main culprit seems to be clause-level evaluation errors, which we have discussed in more detail in Appendix A.4, and mentioned in the original submission.

| Task Name | Fewshot | Neurosymbolic | Logitext |
|---|---|---|---|
| cuad_warranty_duration | 0.50 | 0.19 | **0.61** |
| diversity_2 | 0.76 | 0.35 | **0.83** |
| learned_hands_housing | 0.50 | 0.34 | **0.60** |
| supply_chain_disclosure_best_practice_verification | 0.58 | 0.02 | **0.59** |
| supply_chain_disclosure_disclosed_certification | **0.72** | 0.57 | 0.33 |

Table 2: Correctness on the TG task for Legalbench (100 samples/task)

| Task Name | Fewshot | Neurosymbolic | Logitext |
|---|---|---|---|
| task021_mctaco_grammatical_logical | 0.41 | 0.44 | **0.50** |
| task022_cosmosqa_passage_inappropriate_binary | 0.78 | 0.69 | **0.80** |
| task108_contextualabusedetection_classification | **0.75** | 0.38 | 0.63 |
| task457_matres_conditional_classification | **0.87** | 0.49 | 0.59 |
| task459_matres_static_classification | 0.57 | 0.56 | **0.69** |

Table 3: Correctness on the TG task for SNLI (50 samples/task)

## A.4 CLAUSE-LEVEL ERROR ANALYSIS

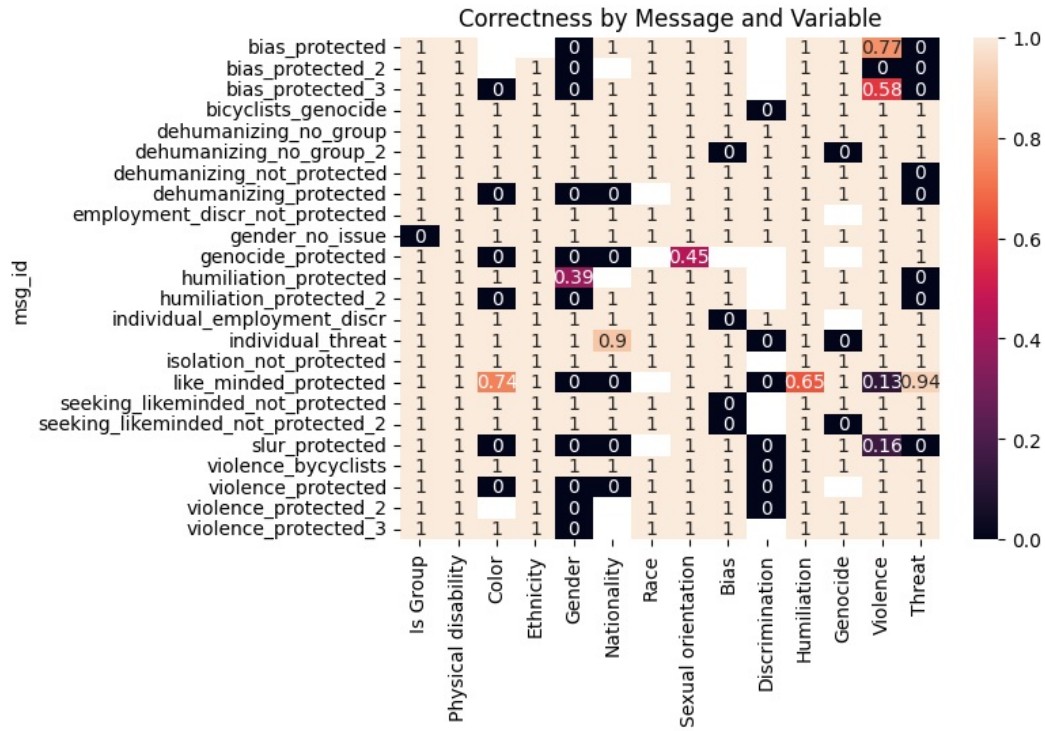

Figure 9: Clause-level accuracy for Disruptive Behavior policy from Figure 1a

Figure 9 is a quick analysis of clause-level accuracy for the "Disruptive Behavior" prompt of Figure 1a. The x-axis of the figure lists the various clauses in the prompt[1]. The y-axis represents individual messages input to the prompt. Each message was run 100 times through the prompt using `gpt-4.1-nano` as the base model. Each entry in the the table is the fraction of times the answer for a particular clause was correct for that message. Some entries are missing because some runs did not complete.

Several points are worth noting, all admittedly only in the context of the current prompt:

1. Sub-clause level inference can be quite stable across runs. They are usually either always correct or always wrong, only occasionally are they not 0 or 1. Thus the worst case of several sub-clauses at a time unpredictably producing errors due to stochastic variation is not inevitable.

2. Inference results are predominantly correct, i.e., sub-clause level accuracy may be much higher than "holistic" document-level accuracy.

3. Certain clauses (i.e., columns, e.g. Gender) are consistently interpreted incorrectly across inputs. In a production setting, we would consider re-wording these, hopefully yielding consistently *correct* clauses.

4. Most inputs (i.e., rows) always have at least one sub-clause evaluated incorrectly. This may seem fatal, until we recall the logic of the prompt is essentially $d =$ isGroup $\land$ (PhysicalDisability $\lor$ Color $\lor$ Ethnicity $\lor$ Gender $\lor$ Nationality $\lor$ Race $\lor$ SexualOrientation) $\land$ (Bias $\lor$ Discrimination $\lor$ Humiliation $\lor$ Genocide $\lor$ Violence). If a clause Gender, which is supposed to be False by default, wrongly evaluates to True, it will not cause an end-to-end error given that Gender is part of a larger disjunction ("or") operation. Thus the precise value of the error, the operation it is part of, and the values of other operands in the operation all contribute to whether a higher-level error is generated. Clausal error does not necessarily imply global error.

While this analysis is by no means comprehensive, it gives some intuition of why the introduction of clause-level errors does not necessarily lead to catastrophic failure at the whole-formula level.

## A.5   Solver algorithms

### A.5.1   Check() algorithm details

The fully detailed version of the Check() algorithm (Figure 10) for solving Logitext constraints is moved here. The version includes details of caching and history, as mentioned in the main body of the paper.

---

[1]Note in the version of the example used in the body of the paper, we omit the Discrimination and Humiliation categories for brevity but they are included in this analysis, which was performed on the complete version of the policy document.

**Algorithm 2** check($D, \pi_D$)

1: **In:** Doc. $D = (vs, us, \phi, \nu)$, asst. $\pi_D$
2: **Out:** UNSAT or satisfying asst. $\pi'_D$
3: **Initialize** logical solver $Z_\phi$ with $\phi$
4: **if** all $u_i \in us$ are bound in $\pi_D$ **then**
5:     **return** LLMVerify($\nu, \phi, \pi_D$)
6: **while** true **do**
7:     //Propose asst. respecting $\phi$ and $\pi_D$
8:     $\pi_Z \leftarrow Z_\phi(vs, \pi_D)$
9:     **return** UNSAT **if** $\pi_Z = \varnothing$
10:     // NLTC solving for unbound text vars
11:     $\pi_s$, satisfiable $\leftarrow \{\}$, true
12:     **for** each unbound $u_j \in us$ **do**
13:         // Use relevant constraints $\mathcal{N}$ for $u_j$
14:         $\mathcal{N} = \{\nu_i \in \nu \mid \nu_i \text{ reads } u_j\}$
15:         $u_j^* \leftarrow$ NLSolver($u_j, \mathcal{N}, \pi_D \cup \pi_s \cup \pi_Z$)
16:         **if** $u_j^*$ is None **then**
17:             $Z_\phi$.block($\pi_Z$)
18:             satisfiable $\leftarrow$ false ; **break**
19:         $\pi_s \leftarrow \pi_s \cup \{u_j = u_j^*\}$
20:     **if not** satisfiable **then continue**
21:     **return** $\pi_s \cup \pi_D \cup \pi_Z$

(a) Outer SMT/NLSolver loop

NLSolver(u, $\mathcal{N}$, $\pi$)

1: **In:** Search target $u$, NLTC set $\mathcal{N}$, its partial asst. $\pi$, Cache $C$, and $T \in \mathbb{N}$
2: **Out:** String value $u^*$ or None
3: **if** $(u, \mathcal{N}, \pi) \in C$ **then**         ▷ Cache Lookup
4:     **return** $C[(u, \mathcal{N}, \pi)]$
5: **else if** $\exists$ $C$.partial_match($u, \mathcal{N}, \pi$) **then**
6:     $u^* \leftarrow C$.closest_partial_match($u, \mathcal{N}, \pi$)
7: **else**
8:     Sample $u^* \sim$ LLM ($\mathcal{N}, \pi, \varnothing, \varnothing, \varnothing$)
9: History $H \leftarrow \{u^*\}$
10: **for** $t = 1$ to $T$ **do**
11:     $sat \leftarrow$ true, $\bar{\pi} \leftarrow \emptyset$, $\tilde{\pi} \leftarrow \emptyset$
12:     **for** $\nu_k \in \mathcal{N}$ **do**
13:         $b_k \leftarrow$ Truth value for $\nu_k$ from $\pi$
14:         $\tilde{b}_k \leftarrow$ LLMVerify($\nu_k, \pi \cup \{u = u^*\}$)
15:         **if** $b_k \neq \tilde{b}_k$ **then**
16:             $sat \leftarrow$ false
17:             $\bar{\pi} \leftarrow \bar{\pi} \cup \{(\nu_k, b_k)\}$
18:             $\tilde{\pi} \leftarrow \tilde{\pi} \cup \{(\nu_k, \tilde{b}_k)\}$
19:     $C[(u, \mathcal{N}, (\pi - \bar{\pi}) \cup \tilde{\pi})] \leftarrow u^*$
20:     **If** $sat$ **then return** $u^*$
21:     $u^* \leftarrow$ LLM($\mathcal{N}, \pi, H, \bar{\pi}, u^*$)
22:     History $H \leftarrow H \cup \{(u^*, \bar{\pi})\}$
23: **return** None

(b) NLSolver: A theory for NLTCs

Figure 10: The check() algorithm for solving logitext constraints

### A.5.2 LLMPropose algorithm details

LLMPropose (Algorithm 3), given a string variable $u$, a set of NLTCs, a partial assignment, produces a text string corresponding to $u$ that satisfies the NLTCs and the partial assignment. LLMPropose is a thin wrapper around an LLM prompt (Appendix A.7.1).

**Algorithm 3** LLMPropose($u$, $N_k$, $\mathcal{P}$)

1: **Input:** string varialbe $u$, NLTCs $N_k$, context value assignments $\mathcal{P}$
2: **Output:** Generated text string
3: $prompt \leftarrow$ Format $N_k$ and $\mathcal{P}$ as a text prompt (Appendix A.7.1) that requests generation of text satisfying $N_k$ given context $\mathcal{P}$
4: $response \leftarrow$ LLM.call($prompt$)
5: $result \leftarrow$ Parse and extract the generated text from $response$
6: **return** $result$

### A.5.3 LLMVerify algorithm details

Given an NLTC and an assignment of variables to values, LLMVerify (Algorithm 4) evaluates the output (Boolean) variable of the NLTC. it is a thin wrapper around the LLM prompt of Appendix A.7.3.

**Algorithm 4** LLMVerify($N_k$, $\mathcal{P}$)

1: **Input:** NL Text constraint $N_k$, context value assignments $\mathcal{P}$
2: **Output:** True/False
3: $prompt \leftarrow$ Format $N_k$ and $\mathcal{P}$ as a text prompt (Appendix A.7.3) that queries whether $N_k$ is True or False based on the context $\mathcal{P}$
4: $response \leftarrow$ LLM.call($prompt$)
5: $result \leftarrow$ Parse the $response$ into True or False
6: **return** $result$

### A.6 Number of LLM calls from NLSolver

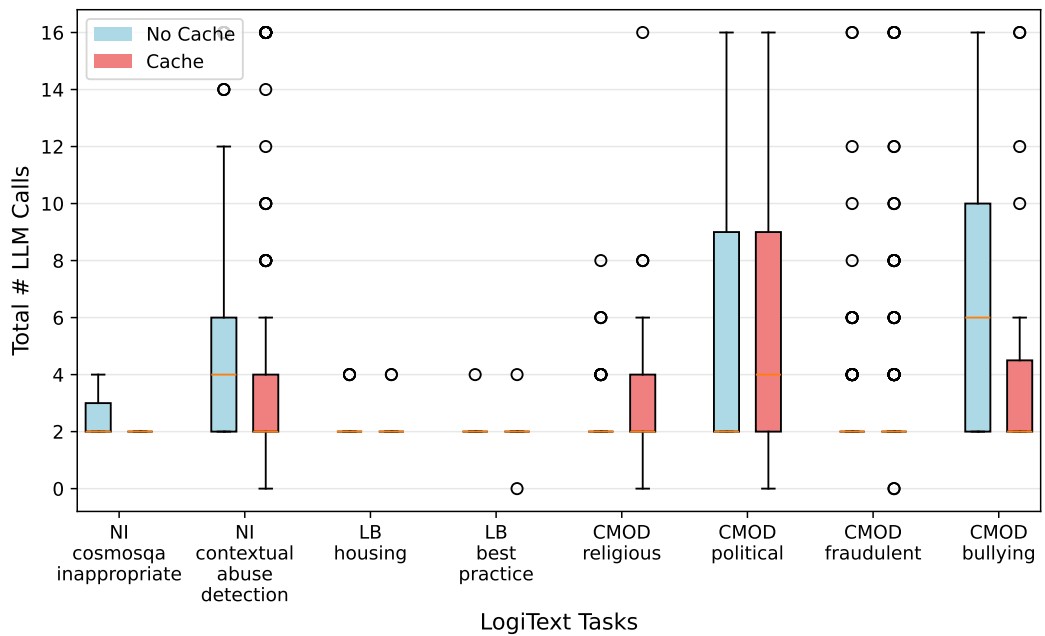

Figure 11: Boxplot of the number of LLM calls per coverage example made by the NLSolver algorithm across all benchmark tasks. The X-axis denotes individual tasks, and the Y-axis reports the distribution of LLM call counts per task.

## A.7 LLM PROMPTS USED FOR NLSOLVER ALGORITHM

These are the LLM prompts for logitext's NLSolver algorithm.

### A.7.1 LLM PROPOSE PROMPT

```
[
    (
        "system",
        "You write concise, clear prose that meets semantic
            constraints. "
        "Return ONLY the text, no preambles."
    ),
    (
        "user",
        """You must produce a text that satisfies the following
            semantic constraint truth values.

Constraints (with target truth values):
{constraints_block}

If some constraints previously failed, focus on satisfying them now:
{failed_block}

Guidelines:
Write one coherent text (few paragraphs max) that attempts to satisfy
    ALL target truth values.
The text should be natural and readable.
Avoid mentioning constraints explicitly.
Do not include JSON. Output ONLY the text.

Previous attempts for context (do NOT copy; improve instead):
{history_block}
"""
```

```
),
]
```

Listing 1: LLM Propose Prompt

### A.7.2 EXAMPLE LLM PROPOSE PROMPT INSTANCE

```
--- System Prompt
You write concise, clear prose that meets semantic constraints. Return
    ONLY the text, no preambles.

--- User Prompt
You must produce a text that satisfies the following semantic
    constraint truth values.
Constraints (with target truth values):
- (C1) target=True :: Content that is a political reference. In this
    case there is
        sufficient context.

        -   **Political Reference** - Words or phrases that are
            political
            references. Includes political titles, entities, slogans,
            government buildings, federal properties, etc.

- (C2) target=False :: Content that is a political reference. In this
    case there is
        sufficient context.

        -   **Political Party or Affiliation** - Name of a political
            party
            or group, including the people that belong to it.

- (C3) target=False :: Content or activity related to politics.

        -   **Activity/Discussion** - Activities or discussions
            related to
            politics. Includes political ideologies, debates,
                campaigning,
            causes, events, etc.

- (C4) target=False :: **Potential Political Reference**
    Content that could be a political reference. In this case it is
        uncertain due to its commonality, multiple meanings, current
            usage,
        and/or insufficient context.

        -   **General Reference** - Words or short phrases that could
            potentially be used in a political manner, although there
                is not
            enough context to make that determination.
- (C5) target=False :: Names of Political figures or people

        For example, "Joe Biden", "Donald Trump".

- (C6) target=False :: References to royal families, their titles and
    duties.
```

```
            For example, "King Charles", "Prince William".

    If some constraints previously failed, focus on satisfying them now:
    - (C2) target=False, predicted=True :: Content that is a political
        reference. In this case there is sufficient context.

            -   **Political Party or Affiliation** - Name of a political
                party
                or group, including the people that belong to it. [why
                    failed last time: No political party names or
                    affiliations appear in the text.]
    - (C3) target=False, predicted=True :: Content or activity related to
        politics.

            -   **Activity/Discussion** - Activities or discussions
                related to politics. Includes political ideologies,
                debates, campaigning, causes, events, etc. [why failed
                last time: The text is about gaming, not politics.]
    - (C5) target=False, predicted=True :: Names of Political figures or
        people

            For example, "Joe Biden", "Donald Trump". [why failed last
                time: No names of political figures appear in the text.]
    - (C6) target=False, predicted=True :: References to royal families,
        their titles and duties.

            For example, "King Charles", "Prince William".  [why failed
                last time: The text does not mention royalty.]

    Guidelines:
    - Write one coherent text (few paragraphs max) that attempts to
        satisfy ALL target truth values.
    - The text should be natural and readable.
    - Avoid mentioning constraints explicitly.
    - Do not include JSON. Output ONLY the text.

    Previous attempts for context (do NOT copy; improve instead):
    That last boss was -insanewe pulled it off just in time. The map\'s
        skyline had a Capitol-like building, which gave a real-world vibe
        without leaving the game.

    ---

    Just got through a brutal boss. The 'maps skyline features a Capitol-
        like dome in the distance, giving the game a real-world vibe
        without leaving the fantasy setting. Came down to a clutch -
        finishtotally worth the grind.
```

Listing 2: LLM Generation Prompt Example for Political CMOD Task

### A.7.3 LLM VERIFY (CONSTRAINT VERIFICATION) PROMPT

```
[
    (
        "system",
        "You are a meticulous verifier. "
        "Given a candidate text and a list of constraints with desired
            truth values, "
        "judge for EACH constraint whether it is semantically TRUE or
            FALSE in the candidate text. "
        "Be strict and literal, not aspirational."
    ),
    (
        "user",
```

```
            """Candidate text:
    \"\"\"
    {candidate}
    \"\"\"

    Evaluate each constraint independently. For each item, return JSON
        array entries of the form:
    {
      "id": "<constraint id>",
      "description": "<verbatim description>",
      "target": true|false,
      "predicted": true|false,
      "rationale": "<short explanation>"
    }

    Constraints:
    {constraints_block}

    Return ONLY valid JSON array, nothing else."""
        ),
    ]
```

Listing 3: LLM Verify Prompt

### A.7.4 EXAMPLE LLM VERIFY PROMPT INSTANCE

```
    --- System Prompt

    You are a meticulous verifier. Given a candidate text and a list of
        constraints with desired truth values, judge for EACH constraint
        whether it is semantically TRUE or FALSE in the candidate text. Be
         strict and literal, not aspirational.

    --- User Prompt

    Candidate text:
    """
    Just cleared a brutal boss. The skyline in the distance features a
        Capitol-like dome, giving the map a real-world vibe while staying
        firmly in fantasy. We pulled off a clutch finish as the timer hit
        zero, grabbed the loot, and exploded in celebration. The 'domes
        presence made the level feel epic without leaning into politics.
    """

    Evaluate each constraint independently. For each item, return JSON
        array entries of the form:
    {
      "id": "<constraint id>",
      "description": "<verbatim description>",
      "target": true|false,
      "predicted": true|false,
      "rationale": "<short explanation>"
    }

    Constraints:
    - (C1) target=True :: Content that is a political reference. In this
        case there is
            sufficient context.
```

```
                -   **Political Reference** - Words or phrases that are
                    political
                    references. Includes political titles, entities, slogans,
                    government buildings, federal properties, etc.

    - (C2) target=False :: Content that is a political reference. In this
        case there is
                sufficient context.

                -   **Political Party or Affiliation** - Name of a political
                    party
                    or group, including the people that belong to it.

    - (C3) target=False :: Content or activity related to politics.

                -   **Activity/Discussion** - Activities or discussions
                    related to
                    politics. Includes political ideologies, debates,
                        campaigning,
                    causes, events, etc.

    - (C4) target=False :: **Potential Political Reference**
        Content that could be a political reference. In this case it is
                uncertain due to its commonality, multiple meanings, current
                    usage,
                and/or insufficient context.

                -   **General Reference** - Words or short phrases that could
                    potentially be used in a political manner, although there
                        is not
                    enough context to make that determination.
    - (C5) target=False :: Names of Political figures or people

                For example, "Joe Biden", "Donald Trump".

    - (C6) target=False :: References to royal families, their titles and
        duties.

                For example, "King Charles", "Prince William".

    Return ONLY valid JSON array, nothing else.
```

Listing 4: LLM Verification Prompt Example for Political CMOD Task

## A.8  LLM PROMPTS USED FOR NEUROSYMBOLIC APPROACH

These are the LLM prompts for the neurosymbolic approach used in Task Execution (TE) experiments.

Decide what level of reasoning is needed for a task, then route to the appropriate reasoning prompt

Routing level 1: LLM-heavy simple decision with minimal Z3 validation

Routing level 2: Boolean logic breakdown with moderate Z3 reasoning

Routing level 3: Complex constraints with heavy Z3 reasoning

### A.8.1  NEUROSYMBOLIC ROUTER PROMPT

---

**Algorithm 5** Neurosymbolic algorithm

---

1: **Input:** Task description
2: **Output:** True/False
3: routing_level ← LLM.call(neurosymbolic_router_prompt.format(task_description)) (Appendix A.8.1)
4: **if** routing_level == 1 **then**
5:     result ← LLM.call(level_one_reasoning_prompt.format(task_description)) (Appendix A.8.2)
6: **else if** routing_level == 2 **then**
7:     result ← LLM.call(level_two_reasoning_prompt.format(task_description)) (Appendix A.8.3)
8: **else**
9:     result ← LLM.call(level_three_reasoning_prompt.format(task_description)) (Appendix A.8.4)
10: **return** result

---

```
[

    ("user",
     """ You are an expert AI judge that analyzes reasoning tasks to
         determine the optimal logical complexity level.

Your job is to route this task directly to the most appropriate
    reasoning level:

LEVEL 1 (Simple Decision): LLM-heavy simple decision with minimal Z3
    validation
- Use for: Simple yes/no questions, straightforward interpretive tasks
- Best when: Single decision path, minimal logical complexity

LEVEL 2 (Propositional Logic): Boolean logic breakdown with moderate
    Z3 reasoning
- Use for: Multiple boolean conditions, AND/OR combinations, decision
    trees
- Best when: Multiple criteria to evaluate, logical paths can be
    separated

LEVEL 3 (First-Order Logic): Complex constraints with heavy Z3
    reasoning
- Use for: Quantifiers, arithmetic, complex relationships, constraint
    satisfaction
- Best when: Numerical calculations, entity relationships,
    mathematical constraints

TASK: {task_description}

ROUTING ANALYSIS:

1. **Task Complexity Assessment:**
    - Does this task involve multiple boolean conditions that can be
        separated? →( Level 2)
    - Does this task involve quantifiers, arithmetic, or complex
        entity relationships? →( Level 3)
    - Is this a simple decision that doesn't need logical breakdown?
        →( Level 1)

2. **Case Factual Richness:**
    - Does the case provide specific numerical values or structured
        data? (supports Level 3)
```

```
      - Does the case have facts for multiple distinct conditions? (
          supports Level 2)
      - Does the case have basic facts for straightforward analysis? (
          supports Level 1)

  3. **Optimal Level Determination:**
      - Level 1: Simple tasks with basic facts
      - Level 2: Multi-condition tasks with sufficient facts for each
          condition
      - Level 3: Complex quantitative tasks with numerical/structured
          data

  Respond in JSON format:
  {{
      "target_level": 1, 2, or 3,
      "reasoning": "Detailed explanation of why this level is optimal",
      "task_complexity": "simple"/"moderate"/"complex",
      "factual_richness": "basic"/"moderate"/"rich",
      "key_indicators": ["list", "of", "specific", "complexity", "
          indicators"]
  }}
  """
      )

  ]
```

Listing 5: Neurosymbolic Router Prompt

### A.8.2 LEVEL 1 REASONING PROMPT

```
  [

      ("user",
      """{z3_syntax_rules}

  PROBLEM: {task_description}

  LEVEL 1 APPROACH - Simple Decision:

  Simple decision uses a single boolean variable to represent the final
      decision.
  Analyze the problem and determine the value of this single decision
      variable.

  STEP-BY-STEP CODE GENERATION:
  1. Import and setup: import z3; s = z3.Solver()
  2. Declare single boolean variable: decision = z3.Bool('decision')
  3. Analyze the problem and determine if decision should be True or
      False
  4. Add constraint: s.add(decision == True) or s.add(decision == False)
  5. Add final constraint: s.add(decision)

  MANDATORY TEMPLATE:
  ```
  import z3
  s = z3.Solver()

  # Single decision variable
  decision = z3.Bool('decision') # add brief description of the decision
      as comment

  # Set decision value based on analysis
  s.add(decision == True)  # or False based on your assignment
```

```
1188    # Final constraint
1189    s.add(decision)
1190    ```
1191
1192    Analyze the problem and determine whether the decision should be True
1193        (YES) or False (NO).
1194
1195    Respond in JSON format:
        {{
1196    "z3_code": "import z3\\ns = z3.Solver()\\ndecision = z3.Bool('decision
1197        ')\\ns.add(decision == True)\\ns.add(decision)",
1198    "assignments": {{
            "decision": {{
1199        "value": true,
1200        "reasoning": "Detailed step-by-step analysis explaining why this
1201            should be True or False"
1202        }}
1203    }}
        }}
1204
1205    CRITICAL:
1206    - Use literal \\n for newlines
1207    - Analyze the problem carefully to determine if decision should be
            True (YES) or False (NO)
1208
1209    - Set decision == True for YES cases, decision == False for NO cases
        - Always end with s.add(decision)
1210    - WARNING: Invalid JSON will cause parsing errors. Double-check
1211        escaping!
1212    """
1213
1214        )
        ]
1215
```

Listing 6: Level 1 Reasoning Prompt

### A.8.3   LEVEL 2 REASONING PROMPT

```
1220    [
1221        ("user",
1222        """{z3_syntax_rules} (Appendix~\ref{app:neurosym-z3-syntax-rules})
1223
1224    PROBLEM: {prompt}
1225
1226    LEVEL 2 APPROACH - Propositional Logic with Systematic Fact Extraction
            :
1227
1228    Propositional logic uses boolean variables and logical connectives (
            AND, OR, NOT).
1229    Break down the problem into boolean conditions and combine them
1230        logically.
1231
1232    SYSTEMATIC FACT EXTRACTION PROCESS:
1233    1. **Identify Boolean Predicates**: Extract all boolean conditions
            from the task description
1234    2. **Map Facts to Predicates**: For each boolean predicate, find
            relevant facts in the case
1235
1236    3. **Evaluate Truth Values**: Carefully assess whether each fact
            satisfies the predicate condition
1237
1238    4. **Cross-Reference Validation**: Verify fact assessments against all
             available case information
1239
1240    5. **Logical Combination**: Combine predicates using appropriate
            boolean operators
1241
        STEP-BY-STEP CODE GENERATION:
```

```
1. Import and setup: import z3; s = z3.Solver()
2. Declare boolean variables (use meaningful variable names): e.g.,
    meaningful_variable_name1 = z3.Bool('meaningful_variable_name1')
3. Create logical combinations: combined = e.g., z3.And(
    meaningful_variable_name1, meaningful_variable_name2)
4. Create final decision: decision = z3.Or(meaningful_path_name1,
    meaningful_path_name2)
5. Add value constraints: s.add(meaningful_variable_name1 == True)
6. Add final constraint: s.add(decision)

FACT EXTRACTION GUIDELINES:
- **Thorough Analysis**: Read the entire case description carefully
    before making assignments
- **Explicit Reasoning**: For each boolean assignment, provide clear
    reasoning based on specific case facts
- **Conservative Assessment**: When facts are ambiguous, err on the
    side of what can be definitively established
- **Context Consideration**: Consider the broader context and
    relationships between different facts
- **Evidence-Based**: Base each boolean value on concrete evidence
    from the case, not assumptions

MANDATORY TEMPLATE:
```
import z3
s = z3.Solver()

# Boolean conditions (extracted from task requirements)
condition_a = z3.Bool('condition_a')
condition_b = z3.Bool('condition_b')
condition_c = z3.Bool('condition_c')

# Logical combinations (reflecting task structure)
primary_path = z3.And(condition_a, condition_b)
alternative_path = condition_c

# Final decision logic
decision = z3.Or(primary_path, alternative_path)

# Value assignments (based on systematic fact extraction)
s.add(condition_a == True)   # Must provide specific case-based
    reasoning
s.add(condition_b == False)  # Must provide specific case-based
    reasoning
s.add(condition_c == True)   # Must provide specific case-based
    reasoning

# Final constraint
s.add(decision)
```

ASSIGNMENT REASONING REQUIREMENTS:
For each boolean assignment in the "assignments" section, you MUST:
1. **Quote Specific Facts**: Reference exact facts from the case
    description
2. **Explain Relationship**: Show how the fact relates to the boolean
    condition
3. **Justify Truth Value**: Clearly explain why the fact makes the
    condition True or False
4. **Consider All Evidence**: Acknowledge any facts that might support
    the opposite conclusion

Respond in JSON format:
{{
```

```
"z3_code": "import z3\\ns = z3.Solver()\\ncondition_1 = z3.Bool('
    condition_1')\\ncondition_2 = z3.Bool('condition_2')\\ndecision =
    z3.And(condition_1, condition_2)\\ns.add(condition_1 == True)\\ns.
    add(condition_2 == False)\\ns.add(decision)",
"assignments": {{
    "condition_1": {{
    "value": true,
    "reasoning": "SPECIFIC case facts that establish this condition as
        true, with explicit quotations and logical connection"
    }},
    "condition_2": {{
    "value": false,
    "reasoning": "SPECIFIC case facts that establish this condition as
        false, with explicit quotations and logical connection"
    }}
}}
}}

CRITICAL REQUIREMENTS:
- Use literal \\n for newlines
- decision must be assigned the logical expression
- Name the final decision variable 'decision'
- Each assignment reasoning must reference SPECIFIC case facts
- Provide detailed evidence-based justification for each boolean value
- Consider the complete case context when making assessments
- WARNING: Invalid JSON will cause parsing errors. Double-check
    escaping!
"""
    )
]
```

Listing 7: Level 2 Reasoning Prompt

### A.8.4 LEVEL 3 REASONING PROMPT

```
[
        ("user",
        """{z3_syntax_rules}(Appendix~\ref{app:neurosym-z3-syntax-
            rules})

PROBLEM: {prompt}

LEVEL 3 APPROACH - First-Order Logic with Systematic Constraint
    Modeling:

CRITICAL JSON SAFETY RULES:
- Double-escape ALL backslashes in z3_code: \\\\ becomes \\\\\\\\
- Double-escape ALL quotes in z3_code: \\" becomes \\\\\\"
- Use \\\\\\\\n for line breaks in z3_code string
- Test your JSON before responding - ensure it's valid

First-order logic includes quantifiers, domain variables, predicates,
    and arithmetic operations.
Systematically model the problem using formal logical constructs and
    constraint relationships.

SYSTEMATIC CONSTRAINT MODELING PROCESS:
1. **Domain Analysis**: Identify entities, values, and relationships
    that need formal modeling
2. **Variable Declaration**: Define appropriate domain variables (Int,
     Real, String, Bool)
3. **Predicate Definition**: Create boolean predicates that capture
    key relationships
```

```
4. **Constraint Formulation**: Build arithmetic and logical
   constraints from requirements
5. **Quantifier Integration**: Add universal/existential quantifiers
   where appropriate
6. **Decision Integration**: Combine all constraints into a unified
   decision formula

FIRST-ORDER LOGIC ELEMENTS:
- Quantifiers: z3.ForAll(), z3.Exists()
- Domain variables: z3.Int(), z3.Real(), z3.String()
- Predicates and relations over domains
- Arithmetic operations: +, -, *, /, >, <, >=, <=
- Complex symbolic reasoning with variables and functions

STEP-BY-STEP CODE GENERATION:
1. Import and setup: import z3; s = z3.Solver()
2. Declare domain variables (use meaningful names): entity = z3.Int('
   entity'); name = z3.String('name')
3. Create predicates: has_property = z3.Bool('has_property')
4. Build arithmetic/comparison expressions: meets_threshold = value >=
    threshold
5. Add quantifiers when needed: z3.ForAll([x], z3.Implies(P(x), Q(x)))
6. Create decision: decision = z3.And(arithmetic_conditions,
   boolean_conditions)
7. Add constraints and final constraint: s.add(decision)

CONSTRAINT MODELING GUIDELINES:
- **Formal Precision**: Use precise mathematical relationships and
    logical operators
- **Complete Modeling**: Capture all relevant constraints and
    relationships from the problem
- **Value Extraction**: Extract specific numerical values, thresholds,
     and measurements from the case
- **Relationship Mapping**: Model complex relationships between
    entities and their properties
- **Quantifier Usage**: Use quantifiers when dealing with universal or
     existential statements

MANDATORY TEMPLATE:
```
import z3
s = z3.Solver()

# Domain variables (extracted from case facts)
entity_value = z3.Int('entity_value')
threshold = z3.Int('threshold')
entity_name = z3.String('entity_name')

# Predicates (boolean conditions from requirements)
has_required_property = z3.Bool('has_required_property')
satisfies_constraints = z3.Bool('satisfies_constraints')

# Arithmetic/comparison expressions (from numerical requirements)
meets_threshold = entity_value >= threshold
value_in_range = z3.And(entity_value >= 0, entity_value <= 1000)

# Quantified expressions (when applicable)
x = z3.Int('x')
universal_property = z3.ForAll([x],
    z3.Implies(x >= threshold, x >= entity_value))

# Combined first-order decision (integrating all constraints)
decision = z3.And(
    meets_threshold,
    has_required_property,
```

```
1404        satisfies_constraints ,
1405        value_in_range ,
1406        universal_property
1407    )
1408
1409    # Value assignments (based on systematic fact extraction)
       s.add(entity_value == 75)
1410    s.add(threshold == 50)
1411    s.add(has_required_property == True)
1412    s.add(satisfies_constraints == True)
1413
1414    # Final constraint
       s.add(decision)
1415    ```
1416
1417    ASSIGNMENT REASONING REQUIREMENTS:
1418    For each variable assignment in the "assignments" section, you MUST:
1419    1. **Value Source**: Clearly identify where each value comes from in
          the case facts
1420    2. **Relationship Explanation**: Explain how the variable relates to
1421       the overall constraint model
1422    3. **Mathematical Justification**: For numerical values, explain the
1423       mathematical reasoning
1424    4. **Constraint Integration**: Show how the variable fits into the
          broader logical framework
1425    5. **Validation Check**: Verify that the assignment is consistent with
1426        all problem requirements
1427
1428    Respond in JSON format:
       {{
1429    "z3_code": "import z3\\ns = z3.Solver()\\nvalue = z3.Int('value')\\
1430       nthreshold = z3.Int('threshold')\\nhas_property = z3.Bool('
1431       has_property')\\nmeets_req = value >= threshold\\nx = z3.Int('x')
1432       \\nuniversal = z3.ForAll([x], z3.Implies(x >= threshold, x >=
1433       value))\\ndecision = z3.And(meets_req, has_property, universal)\\
1434       ns.add(value == 75)\\ns.add(threshold == 50)\\ns.add(has_property
          == True)\\ns.add(decision)",
1435    "assignments": {{
1436        "value": {{
1437        "value": 75,
1438        "reasoning": "Value source: [specific case fact]. Relationship: [
            how it relates to constraint model]. Mathematical
1439            justification: [numerical reasoning]. Constraint integration:
1440            [role in decision formula]."
1441        }},
1442        "threshold": {{
1443        "value": 50,
            "reasoning": "Value source: [specific case fact]. Relationship: [
1444            how it relates to constraint model]. Mathematical
1445            justification: [numerical reasoning]. Constraint integration:
1446            [role in decision formula]."
1447        }},
1448        "has_property": {{
1449        "value": true,
            "reasoning": "Value source: [specific case fact]. Relationship: [
1450            how it relates to constraint model]. Boolean justification: [
1451            why True/False]. Constraint integration: [role in decision
1452            formula]."
1453        }}
1454    }}
       }}
1455
1456    CRITICAL REQUIREMENTS:
1457    - Use literal \\n for newlines
```

```
1458         - Must include first-order logic elements (quantifiers, domain
1459            variables, arithmetic)
1460      - decision must be assigned the complete logical expression
1461      - Name the final decision variable 'decision'
1462      - Each assignment reasoning must follow the structured format above
1463      - Systematically extract and model all relevant numerical and
1464            structural data
            - Use appropriate mathematical and logical operators for constraint
1465            relationships
1466      - WARNING: Invalid JSON will cause parsing errors. Double-check
1467            escaping!
1468      """
              )
1469      ]
1470
```

Listing 8: Level 3 Reasoning Prompt

### A.8.5  Z3 SYNTAX RULES

```
*********** Z3 SYNTAX RULES (Must Follow Exactly): ***********

0. INDENTATION RULES (CRITICAL - PREVENTS "unexpected indent" ERRORS):
- Use EXACTLY 4 spaces for each indentation level
- NO TABS allowed - only spaces
- All lines at same level must have identical indentation
- Check each line starts with correct number of spaces
- WRONG: Mixed spaces/tabs cause "unexpected indent" errors

1. BOOLEAN OPERATIONS:
- CORRECT: z3.And(var1, var2, var3)
- CORRECT: z3.Or(var1, var2)
- CORRECT: z3.Not(var1)
- CORRECT: z3.Implies(var1, var2)
- WRONG: var1 and var2  (Python operators don't work in Z3)
- WRONG: var1 or var2
- WRONG: not var1

2. CONSTRAINT ASSIGNMENT:
- CORRECT: s.add(variable == True)
- CORRECT: s.add(variable == False)
- CORRECT: s.add(variable == z3.And(cond1, cond2))
- WRONG: s.add(variable == (cond1 and cond2))
- WRONG: variable = cond1 and cond2

3. BOOLEAN VARIABLES ONLY IN Z3 FUNCTIONS:
-  CORRECT: z3.And(bool_var1, bool_var2)
-  CORRECT: z3.Or(condition_a, condition_b)
-  WRONG: z3.And('text', 'text')  (String literals cause errors)
-  WRONG: z3.Or('name1', 'name2')

4. FINAL DECISION CONSTRAINT:
-  ALWAYS END WITH: s.add(decision)
-  NEVER: if s.check() == sat: ...
-  NEVER: print(s.model())

5. VARIABLE DECLARATIONS:
-  CORRECT: var_name = z3.Bool('var_name')
-  CORRECT: amount = z3.Int('amount')
-  CORRECT: rate = z3.Real('rate')

6. DECISION VARIABLE ASSIGNMENT:
- CORRECT: decision = z3.And(condition1, condition2)
- CORRECT: decision = z3.Or(path1, path2)
```

```
     - WRONG: decision = z3.Bool("decision")   (Should avoid creating
        unrelated variable)

   7. PYTHON LIST VS Z3 ARRAY DISTINCTION (CRITICAL):
   - NEVER mix Python list indexing with Z3 symbolic variables
   - WRONG: python_list[z3_variable] # Causes "list indices MUST be
        integers" error
   - CORRECT: Use Z3 arrays: z3.Array('array_name', z3.IntSort(), z3.
        BoolSort())
   - CORRECT: Use explicit variables: var1, var2, var3 for small fixed
        sets

   8. COMMON ERROR PATTERNS TO AVOID:
   - Don't access Python lists/arrays with Z3 symbolic variables
   - Don't mix Python boolean operators (and/or/not) with Z3 expressions
   - Don't create unnecessary intermediate Boolean variables when direct
        expressions work
   - Don't use Python string comparison with Z3 string variables
```

Listing 9: Z3 Syntax Rules for above prompts

## A.9    CONTENT MODERATION DATASET EXAMPLE

### A.9.1    FRAUDULENT CONTENT POLICY

```
   ```@python(input_message)(result_var)

   ```
   # Fraudulent

   ## Topic Definition

   The Fraudulent Topic is used to identify attempts to deceive a victim
   into providing funds or private information.

   ## Critical Information

   -    Online fraud refers to online content and activity that uses
        misrepresentation to deceive a victim into providing funds or
        private information. Misrepresentation is often accomplished by
        impersonation.

        -    Impersonation is where an individual falsely claims to be, or
             presents themselves to be, another real or fictional
                 individual,
             group, label, or entity.

   -    One of the most common methods used to commit online fraud is
        phishing.

        -    **Phishing** is the fraudulent practice of sending emails or
             other messages claiming to be from reputable companies in
                 order
             to induce individuals to reveal personal information (e.g.,
             passwords or credit card numbers).

   -    Online fraud includes but is not limited to:

        -    **Consumer investment fraud**

             -    The expected benefit is investment returns and includes
                  fake
                  shares, Ponzi schemes, film frauds, etc.
```

```
        -    **Consumer products and services fraud**

             -    The expected benefit is the product or service and this
                  includes fake tickets, bogus holidays, dietary pills that
                  don't work, products that don't arrive, etc.

        -    **Employment fraud**

             -    The expected benefit is employment and these include fake
                  opportunities for jobs such as work at home scams, model
                  agency work, etc.

        -    **Prize and grant fraud**

             -    The expected benefit is winning a prize or other windfall
                  and this includes fake lotteries, 419 scams (e.g.,
                      Nigerian
                  prince), etc.

        -    **Phantom debt collection fraud**

             -    The expected benefit is avoiding the consequences of
                  failing
                  to pay debts the victim did not know were previously owed
                  and this includes bogus demands for payment for debts,
                  taxes, etc.

        -    **Charity fraud**

             -    The expected benefit is contributing to a charity, but the
                  reality is that the victim is contributing to the
                  fraudsters, not a legitimate cause.

        -    **Relationship and trust fraud**

             -    The expected benefit is a relationship, but the reality is
                  usually a fake identity aimed at securing monies from the
                  victim.

        -    **Identity Fraud**

             -    Personal data is extracted from the victim or from a third
                  party (such as the victim's bank).

    -    Currently, Community Sift does not provide a complete solution for
         this Topic!

    -    Sift operates on single lines of text. This is a complex, nuanced,
         and context-heavy Topic.

    -    For now, we are considering these as future expansions as we add
         more context capabilities to our product.

    -    old subtopics

         -    Hacking References

              -    References to hacking accounts, games, or similar.

         -    Account Fraud

              -    Selling, exchanging, swapping, or advertising accounts,
                   account information, currency, or similar.
```

```
        -    Phishing Attempts

            -    Attempts to scam or induce individuals to reveal personal
                 information (e.g., account details, passwords, financial
                 information, etc.) for fraudulent purposes.

    ## Subtopics & Subcategories

    -    **Hacking and Cheating**\
         {{ let matches_hacking_cheating_subcategory = [[ Content and
             activity that uses, shares, or promotes illegal ways of
         obtaining currency, memberships, or similar in-app perks or
         resources in gaming and/or social accounts.
         {{forsome

         [[-    **Hacking Reference** - Content or activity that references
             in-app hacking.
         ]]

         [[-    **Cheating Reference** - Content or activity that references
             in-app cheating.
         ]]
         }}]]   where "content" is input_message and "activity" is
             input_message }}
    -    **Online Fraud**\
         {{ let matches_online_fraud_subcategory = [[Content and activity
             that uses misrepresentation to deceive a victim
         into providing funds or private information.
         {{forsome
    [[
         -    **Account and Password Fraud** - Content or activity that
             attempts to request or facilitate the sharing, stealing,
                 buying,
             or exchanging of in-app accounts or passwords.
    ]]

    [[
         -    **Consumer Investment Fraud** - Content or activity where the
             expected benefit is investment returns and includes fake
                 shares,
             Ponzi schemes, film frauds, etc.
    ]]

    [[
         -    **Consumer Products and Services Fraud** - Content or activity
             where the expected benefit is the product or service and this
             includes fake tickets, bogus holidays, dietary pills that don'
                 t
             work, products that don't arrive, etc.
    ]]

    [[
         -    **Employment Fraud** - Content or activity where the expected
             benefit is employment and these include fake opportunities for
             jobs such as work at home scams, model agency work, etc.
    ]]

    [[
         -    **Prize and Grant Fraud** - Content or activity where the
             expected benefit is winning a prize or other windfall and this
             includes fake lotteries, 419 scams (e.g., Nigerian prince),
                 etc.
    ]]

    [[
```

```
    -   **Phantom Debt Collection Fraud** - Content or activity where
        the expected benefit is avoiding the consequences of failing
            to
        pay debts the victim did not know were previously owed and
            this
        includes bogus demands for payment for debts, taxes, etc.
]]

[[
    -   **Charity Fraud** - Content or activity where the expected
        benefit is contributing to a charity, but the reality is that
        the victim is contributing to the fraudsters, not a legitimate
        cause.
]]

[[
    -   **Relationship and Trust Fraud** - Content or activity where
        the
        expected benefit is a relationship, but the reality is usually
            a
        fake identity aimed at securing monies from the victim.
]]

[[
    -   **Identity Fraud** - Content or activity where the personal
        data
        is extracted from the victim or from a third party (such as
            the
        victim's bank).
]]
     }} ]] where "content" is input_message and "activity" is
         input_message }}
## Exclusions

The message does NOT match the topic if any of the following subtopics
AND their respective subcategories hold:

-   **Legitimate Codes**\
    {{let exclusion_legitimate_codes = [[
    Content or activity related to legitimate "cheat" codes or
    promotional ("promo") codes that are part of the app or game.

    -   **Legitimate Codes**\
        Content or activity related to legitimate "cheat" codes or
        promotional ("promo") codes that are part of the app or game.,
        e.g., \"enter this promo code \[CODE\] to redeem 3 booster
        packs!\"
    ]] where "content" is input_message and "activity" is
        input_message }}
## Topic Overlaps

The Fraud Topic can overlap with the following Topics:

-   **PII**

    -   When the fraudulent content includes personally identifiable
        information.

## Risk Level Mapping

If the message matches a topic AND a subcategory under a risk level
    AND
the intensity of the sentiment matches the examples, it matches that
topic at that risk level
```

```
-    **Hacking and Cheating**

    -    LOW

        -    Hacking Reference, e.g., \"it's for hacking accounts\", \"
             i
             will hack your account\"

        -    Cheating Reference, e.g., \"use the coin cheat to
             duplicate
             all your gold\"

-    **Online Fraud**

    -    MEDIUM

        -    Account and Password Fraud, e.g., \"anyone looking to
             trade
             accounts?\"

        -    Consumer Investment Fraud, e.g., \"hey come sell herbalife
             with me\"

        -    Consumer Products and Services Fraud, e.g., \"you should
             try
             \[NAME\]! They are miracle weight loss pills!\"

        -    Employment Fraud, e.g., \"work from home and make millions
             just playing games!\"

        -    Prize and Grant Fraud, e.g., \"you're the 100,000 visitor!
             You're a winner! Follow these instructions to claim your
             prize!\"

        -    Phantom Debt Collection Fraud, e.g., \"our records
             indicate
             you have an outstanding payment of \451.35 dollars -
             contact our offices asap\"

        -    Charity Fraud, e.g., \"this xmas, donate to those who need
             it more at \[FRAUDULENT CHARITY\]\"

        -    Relationship and Trust Fraud, e.g., \"young guy looking
             for
             love, prefer to date rich older women\"

        -    Identity Fraud, e.g., \"example\"

    -    HIGH

        -    Account and Password Fraud, e.g., \"after you send the
             money
             i will give you my account info and password so you can
                 log
             in\"

        -    Consumer Investment Fraud, e.g., \"provide us with you
             financial information and a cheque for \[AMOUNT\] so we
                 can
             add you to our system and immediately double your money!
             you'll get rich in no time\"

        -    Consumer Products and Services Fraud, e.g., \"click this
             link \[LINK\] to get your ultra exclusive \[PRODUCT\] at
                 1/2
```

```
                         the price!\"

           -     Employment Fraud, e.g., \"send us your bank information
                 and
                 credit card information to receive your amazing paycheques
                 from \[FAKE COMPANY\]!\"

           -     Prize and Grant Fraud, e.g., \"click this private and
                 secure
                 link to include your credit card information and claim
                     your
                 million dollars!! offer expires today!\"

           -     Phantom Debt Collection Fraud, e.g., \"The authorities are
                 on their way regarding your \[CRIME\]. Settle payment now
                     to
                 prevent this. We accept all credit cards\"

           -     Charity Fraud, e.g., \"we need your contact information
                 and
                 credit card information to send you a tax credit for your
                 donation!\"

           -     Relationship and Trust Fraud, e.g., \"I need \[QUANTITY\]
                 dollars for my plane ticket to come meet you baby. You're
                 the love of my life and we'll get married soon. Just need
                 more money to make it there.\"

           -     Identity Fraud, e.g., \"i used this program \[NAME\] and
                 with the pictures, voice, and personal information of
                 \[PERSON\] you're able to get their bank to confirm their
                 details\"

    ## Questions/Concerns

    -    N/A

    Task: The above document describes a particular policy that we want to
         check whether an input message violates. Your task is to evaluate
         whether a given input message violates the above policy. The
        message violates the above policy if it matches one or more
        subtopics and no exclusions.
    ```@python
    matches_any_exclusions = exclusion_legitimate_codes
    matches_fraud_subtopics = matches_hacking_cheating_subcategory or
        matches_online_fraud_subcategory
    fraud_policy_checker = matches_fraud_subtopics and (not
        matches_any_exclusions)

    result_var = fraud_policy_checker
    ```
```

Listing 10: Fraudulent v.lt

