# OpenReview forum: "Neurosymbolic Language Reasoning as Satisfiability Modulo Theory"
_ICLR.cc/2026/Conference — Submitted to ICLR 2026_

### Official Review · Reviewer_WyQz · 2025-10-21

**Soundness:** 4
**Presentation:** 3
**Contribution:** 4
**Rating:** 8
**Confidence:** 5

**Summary:**

The paper describes an approach to the logical annotation of unstructured text documents that allows the definition of logical constraints in natural language in the context of a semi-structured prompt to a hybrid LLM/SMT solver. This allows the document to be translated into an SMT theory, formed by combining LLM valuations for atomic propositions within the constraints with auto-formalization of complex statements in the document, for which a solver then finds a satisfying assignment or determines the theory is unsatisfiable.

**Strengths:**

An effective and innovative approach to logical reasoning with LLMs that uses an SMT solver as cognitive scaffolding. This is an improvement over previous approaches which depend on autoformalization into logical statements followed by independent execution of a solver over the generated theory. The deeper integration of the LLM into the core of a hybrid reasoning engine is an important direction to pursue, because it begins to address how best to exploit the complementary strengths of formal reasoners and the approximate retrieval of LLM parametric knowledge for reasoning. The presentation is clear and for this reviewer informative and thought-provoking, the description of the formalization for natural language text constraints and the NLSolver were thorough easy to follow. The author(s) effectively make the case that using an LLM to ground atomic statements in the context of a solver is a viable, practical approach to reasoning in neurosymbolic systems that addresses some of the shortcomings of current approaches.

**Weaknesses:**

The paper extends SMT with a theory for textual constraints, but does not discuss formal soundness or completeness guarantees. What are the conditions under which the NLSolver is guaranteed to find a solution if one exists? The paper would benefit from a more rigorous theoretical treatment of these questions.

The paper doesn't adequately address how it ensures factuality of LLM evaluations when these serve as atomic propositions in the logical theory. What happens when the LLM misclassifies a natural language constraint? The LLMVerify function is treated as an oracle, but in practice, LLMs can be inconsistent or incorrect. The paper would benefit from error analysis on how LLM evaluation errors propagate through the logical reasoning process and what safeguards exist to detect or mitigate such errors.

**Questions:**

Have the authors investigated the usability of the Logitext language from a human factors perspective? What is the learning curve for non-experts to write effective annotations? How error-prone is the annotation process? Understanding the practical challenges of getting users to correctly annotate documents is crucial for real-world deployment.

How does the system handle ambiguous natural language that could have multiple valid autoformalizations? Does it maintain multiple hypotheses or commit to a single interpretation? This is particularly important for legal and policy documents where ambiguity may be intentional or unavoidable.

Can the authors provide any formal guarantees about their system? For example, under what conditions is the NLSolver guaranteed to be sound, never returning an incorrect solution? Even partial guarantees would strengthen the theoretical contribution.

---

> ### Author Response · Authors · 2025-11-20
>
> We thank the reviewer for these thoughtful comments. The points raised overlap with broader issues discussed in our overall summary, to which we respectfully refer when appropriate. Below we address each question concisely.
>
> > Q1 & Q5. Formal guarantees of NLSolver (soundness, completeness, correctness).
>
> We agree that these are serious gaps today. There are two reasons why these guarantees are hard. First, regarding completeness, given that the underlying constraint solving problem is at least NP Hard, it is almost inevitable that any practical algorithm will time out while searching for a solution, as theorem provers do today. Of course, we can still ask the question whether if left to run indefinitely Logitext will find a solution. Second, regarding soundness, it is unclear what guarantees we can give since we rely on the LLM for our solutions, and LLMs famously come with no guarantees. Having said all that, we agree that it is a deep and important problem to understand what (e.g., even statistical) guarantees we can give.
>
> > Q2. Factual reliability of LLM evaluations and error propagation.
>
> We respectfully refer the reviewer to Issue S2 in the overall summary, which directly addresses this concern. Although we have no formal guarantees, the empirical question of why noise in evaluation subclauses is not catastrophic with respect to overall accuracy is a critical one for Logitext-type systems where statistical and discrete techniques interface.
>
> > Q3. Usability and annotation difficulty.
>
> We refer the reviewer to Issue S1, which discusses annotation effort and practicality.In a nutshell, Logitext annotation is well within the "prompt engineering" overhead of getting prompts ready for deployment. Based on our experience annotating multi-page prompts such as those in CMOD, the process is manageable once the syntax (e.g., `let` bindings) is understood. While we have not yet conducted a formal user study, we plan to evaluate usability and annotation consistency more systematically in future work.
>
> > Q4. Handling ambiguous or multiply-interpretable text.
>
> As discussed in Issue S5, Logitext currently resolves ambiguity by selecting a single partial formalization when multiple decompositions are possible. This design keeps reasoning deterministic, but we acknowledge that supporting multiple candidate formalizations could be valuable, especially for legal or policy documents where ambiguity is inherent, and note this as a potential extension.

---

### Official Review · Reviewer_AF7g · 2025-10-22

**Soundness:** 2
**Presentation:** 1
**Contribution:** 3
**Rating:** 2
**Confidence:** 4

**Summary:**

This paper proposes Logitext, a neuro-symbolic language that bridges the gap between natural language and formal language. Logitext is represented as natural language text constraints, which streamlines a logical structure of the natural language text. Furthermore, authors extend an SMT solver and propose a NLSolver using an LLM to solve natural language text constraints. The proposed method is evaluated on 15 tasks and yield impressive performance on three types of settings, supporting the effectiveness of the method.

**Strengths:**

1. The paper newly introduces a language called Logitext to fully represent the logical structure behind the natural language text. This is a very important problem in legal, political, and societal domain.
2. The performance is impressive. Logitext performs much better than few-shot prompting and the previous neurosymbolic approach on various benchmarks.

**Weaknesses:**

The presentation should be further improved for clarifications.

### Major points
- There are lots of words "solver", but I'm confused which solver do you exactly mean. I think you'd better come up with a clearer notation. There are lots of related words (e.g., solver, logical solver, SMT solver, LLM-based solver, symbolic solver), but for people who read this paper for the first time, it's really hard to get which "solver" do you mean in the paper throughout.
- For Logitext, what if an implicit premise is hidden in the text, so that you can't capture the entire underlying logical structure of the text by simply annotating it?
- The description of algorithm (Section 3.3) could be further improved. Now, there are too many variables only defined in Figure 4, not in the main paper, which hinders to fully understand the details.
- For the new benchmark CMOD, what is the source of these moderation policies? I cannot find it in anywhere.
- Line 367: 10+ instances per task seems so small. Elaborate the dataset size for each task.
- Line 425-426: Authors should elaborate how this neurosymbolic prompt looks like. If it is something from a previous work, then authors should cite that work.

### Minor points
- In Figure 2 (b), authors should mention that the definition of combinatorial gap is in Appendix A.1. Readers would be confused.
- Which solver did you use in Section 2.2? Did you use NLSolver or an SMT solver?
- In Figure 3, why is there C8? for defining disruptive behavior and immediate threat, you even do not use C8, and as the definition of C6 shows, i guess C8 is equivalent to C6.
- Line 213, if I understand correctly, the first `<var>` and the second `<var>` point two different ones. for clarification, you should indicate those as `<var1>` and `<var2>` instead. Also, what do `<var>`, `<clause>`, and `<phrase>` exactly mean? Authors should elaborate this right after they describe the whole new notations.
- Line 217, authors should cite related papers for `pyz3`.
- Line 253-254, authors should describe how u_i is related to c_jh and what do p_i's mean here. I could understand from the context, but to formally define Logitext, authors should consider this. Also, why does this formalization re-occur here? I think it's already briefly describe in 3.1 Clause naming.
- Line 263, the notation l_jh suddenly appears. what does it mean?
- Line 268: in the previous paragraph, NLTC was notated as v_jh but here it is notated as v_i where 1<=i<=n. Could authors clarify this point?
- Line 412: Text is overlapped by Figure 8.

**Questions:**

See Weaknesses.

---

> ### Author Response · Authors · 2025-11-20
>
> We sincerely thank the reviewer for the detailed comments and helpful suggestions on improving clarity and presentation. Many of these points concern readability and notation, and we have made substantial revisions accordingly. Some issues overlap with those already discussed in the overall summary, to which we respectfully refer the reviewer where appropriate.
>
> ### Major comments
>
> > There are many occurrences of “solver,” but it is unclear which solver is meant (e.g., solver, logical solver, SMT solver, LLM-based solver, symbolic solver).
>
> We fully agree that this terminology was confusing. In the revision we have standardized the notation and now clearly distinguish among the three solver types used in the paper:
> 1. LLM-based solver: the language model performing sub-clause evaluation;
> 2. SMT solver: the symbolic engine (Z3) used for constraint reasoning; and
> 3. NLSolver: our neurosymbolic inference procedure combining the two.
>
> Specifically, we precede every mention of a "solver" by a clarification of which solver we are talking about.
>
> > What if an implicit premise is hidden in the text and cannot be captured by annotation?
>
> We respectfully refer the reviewer to Issue S5 in the overall summary, which discusses precisely this limitation. Logitext is designed to handle such cases by interpreting each annotated clause in the context of the entire document. When essential premises remain implicit, the framework simply reverts to standard LLM prompting for that portion rather than producing an incorrect formalization.
>
> > Section 3.3's algorithm description could be clearer; too many variables appear only in Figure 4.
>
> We appreciate this observation. As noted in Issue S3, we have completely rewritten all of Section 3 to improve clarity and accessibility. In particular, we moved the detailed algorithm description of Section 3.3 to the appendix, and provide a simplified version that gets at the essence of the solution. We then provide a much more detailed description that touches on every line of the new algorithm.
>
> > For the new benchmark CMOD, what is the source of the moderation policies?
>
> They are slightly altered and anonymized versions of an internal moderation policy for user-generated short text from a large production service.
>
> > Line 367: “10+ instances per task” seems small. Elaborate dataset size.
>
> We refer the reviewer to Issue S4, where we describe the expanded evaluation. The dataset now contains 100 instances per task, and our early evaluation shows that results remain qualitatively the same.
>
> > Lines 425-426: please elaborate how the neurosymbolic prompt looks; cite if from previous work.
>
> We describe this prompt in detail in Appendix A.8. It is not directly based on previous work, but it dynamically composes code for the problem and then executes it, a common pattern. Our variant is somewhat more sophisticated, as described in A.8.
>
> ### Minor comments
>
> > Figure 2 (b): mention that the definition of the combinatorial gap is in Appendix A.1.
>
> We have added this pointer below the figure caption to prevent confusion.
>
> > Which solver was used in Section 2.2: NLSolver or SMT solver?
>
> We now explicitly state that ‘a solver’ in the Section 2.2 refers to the SMT solver (Z3) for generating constraint satisfying assignments, not NLSolver.
>
> > In Figure 3, why is there C8?
>
> C8 is indeed redundant. We mainly introduced it to show that variables that are defined in code blocks may be used in subsequent text blocks. However, we agree this is a weak example.
>
> > Line 213: clarify `<var>` notation; distinguish `<var1>`, `<var2>`; define `<var>`, `<clause>`, `<phrase>`.
>
> We have rewritten this section heavily to provide much more explanation of the notation, including with examples.
>
> > Line 217: cite related papers for pyz3.
>
> A proper citation to the pyz3 library documentation and related publications has been added.
>
> > Lines 253–254: clarify how uᵢ relates to $c_{j,h}$ and $p_i$; avoid repeated formalization.
>
> In the revised paper, we clarified the relationships among these symbols to be consistent with the former section’s notations.
>
> > Line 263: $I_{j,h}$ notation appears suddenly.
>
> The subscripts have now be removed.
>
> > Line 268: inconsistent notation ($v_{j,h}$ vs $v_i$).
>
> The new simplified subscripts avoid this.
>
> > Line 412: figure overlaps text.
>
> We corrected the layout in the new draft to ensure proper alignment and spacing.

---

> > ### Comment · Reviewer_AF7g · 2025-11-24
> >
> > Thanks for the effort! I can see that there are huge updates. Here are my follow-up responses.
> >
> > > We respectfully refer the reviewer to Issue S5 in the overall summary, which discusses precisely this limitation. Logitext is designed to handle such cases by interpreting each annotated clause in the context of the entire document. When essential premises remain implicit, the framework simply reverts to standard LLM prompting for that portion rather than producing an incorrect formalization.
> >
> > I’m still not persuaded by the authors’ justifications. For me, as authors said, if Logitext’s contribution is: formalize what we can clearly formalize and not formalize what we cannot (or hard to) formalize, then I cannot see a clear benefits of using this language. Of course, it could clarify some formalizable NL, but then it feels like NL with some annotations that partially specify logical relations of the text. This incomplete language is not intriguing, at least for me. If the authors really try to bridge the gap between natural language and formal language, I think implicit premises that are not clearly present in the text should also be reconstructed and formalized. Although there are some levels of uncertainty and ambiguity as authors said, simply ignore these and keep these as an original NL makes me not interesting.
> >
> > > We appreciate this observation. As noted in Issue S3, we have completely rewritten all of Section 3 to improve clarity and accessibility. In particular, we moved the detailed algorithm description of Section 3.3 to the appendix, and provide a simplified version that gets at the essence of the solution. We then provide a much more detailed description that touches on every line of the new algorithm.
> >
> > If anything is changed, especially if the change is large, then please mark as blue or other colors so that reviewers could clearly see what are changed. Since authors changed the manuscript by a non-trival amount as they mention, I’ll take a closer look at each part of the updated draft after the authors do that annotation job.
> >
> > > Section 3.3's algorithm description could be clearer; too many variables appear only in Figure 4.
> >
> > Thanks for the detailed explanation, but what is the exact source? “An internal moderation policy from a large production service” is too vague for me.

---

> > > ### Author Response · Authors · 2025-11-27
> > >
> > > Thank you again for the careful reading and for acknowledging the extent of the revisions. We address your follow-up comments below.
> > >
> > >
> > > > … If the authors really try to bridge the gap between natural language and formal language, I think implicit premises that are not clearly present in the text should also be reconstructed and formalized.
> > >
> > > We would like to emphasize that we consider the in-context evaluation of predicates to be a strength of Logitext. The context (i.e. the text surrounding the "let" clauses) often includes implicit premises.
> > >
> > > For instance, consider a policy document titled "Privacy Policy for Mobile Device Data". The body of the document may contain the clause requiring that "the stored data should not include sensitive information". Although the phrase "sensitive information" is very broad, the "implicit premise" from the title (above) narrows it to privacy sensitivity.
> > >
> > > Logitext is naturally able to handle such context because NLTCs include the original document d as part of their definition and the clause in the NLTC is evaluated in the context of the corresponding document.
> > >
> > > If the reviewer has other types of implicit premises in mind, we would be grateful if they share an example or more detail.
> > >
> > > > If anything is changed, especially if the change is large, then please mark as blue or other colors so that reviewers could clearly see what are changed.
> > >
> > > Thank you for the suggestion. We have highlighted all substantive changes from the original submission in color and included the annotated PDF inside the **supplementary material ZIP file**. We avoided overwriting the main document because the change markings can sometimes be distracting. We hope this will make it easier to review the revisions.
> > >
> > > > Thanks for the detailed explanation, but what is the exact source?
> > >
> > > We understand the request for more specificity. Due to the double-blind review process, we cannot disclose the exact institution, product, or service name without compromising anonymity. What we can state is that the policy is from a globally deployed, billion-scale messaging service currently in production. We hope this level of detail is acceptable within the anonymization constraints.

---

### Official Review · Reviewer_EsVD · 2025-10-30

**Soundness:** 2
**Presentation:** 2
**Contribution:** 1
**Rating:** 2
**Confidence:** 3

**Summary:**

The authors introduce Logitext - which is a neurosymbolic language to extend NLTCs to SMT solvers. They identify "compositional" and "combinatorial" reasoning gaps in LLMs when dealing with certain types of documents. In these cases, Logitext lets you formalize part of the document (the "logic") - and then uses an iterative solver with Z3 for logical and an NLSolver for textual constraints.

However, this method feels quite contrived, and I don't see any applicability of this beyond some carefully curated examples that require a lot of manual annotation anyways. The complexity of the Logitext systems appears to not be proportionate to the demonstrated gains.

**Strengths:**

1. The paper does address a real problem wrt LLMs struggling with logical consistency in policy documents.
2. The paper is clearly motivated through empirics on reasoning gaps.
3. To my knowledge, this mixture of SMT solvers with NL constraints is novel.

**Weaknesses:**

1. The Logitext system seems quite contrived and requires significant manual effort to convert natural docs.
2. Manual annotation of the logical structure defeats the purpose of scalable automated reasoning. Therefore real usage of this is highly questionable.
3. The convergence of the NLSolver is not guaranteed and the caching employed seems quite ad-hoc.
4. Some of the proposed baselines appear weak and raise concerns about high quality evals.

**Questions:**

1. How does the cost of annotation simply compare to using better prompting strategies?
2. How sensitive is the performance to annotation quality and completeness.
3. What happens (as is likely) when logical structures in the document are incomplete or vague?

---

> ### Author Response · Authors · 2025-11-20
>
> Thank you for your careful reading and thoughtful comments. Several of your points overlap with broader issues raised by other reviewers. When that is the case, we respectfully refer to the corresponding discussion in our overall summary and provide clarifications specific to your review below.
>
> > Logitext is contrived and requires significant manual effort to convert natural docs.
>
> We respectfully refer the reviewer to Issue S1 of our overall summary, which discusses the practical overhead of annotation and its alignment with deployment-stage prompt engineering, which is indeed manual but still broadly necessary.
>
> > Manual annotation defeats the purpose of scalable automated reasoning.
>
> This concern is addressed in Issues S1 and S5. Logitext does not aim to fully formalize entire documents, which would indeed be impractical. Rather, it supports partial formalization, expressing what can be clearly structured while leaving the rest as natural text, allowing reasoning to scale to realistic, messy inputs.
>
> > The convergence of the NLSolver is not guaranteed and the caching seems ad-hoc.
>
> This is indeed true. NLSolver is a best-effort algorithm. In that sense, however, it is similar to all logical solvers: since the underlying problems are NP hard, all solvers today time out on complex problem and therefore cannot guarantee a solution.
>
> Partly based on your feedback, we have reduced focus on caching. As part of our complete rewrite of Section 3 (please see Issue s3), we have moved caching and related discussion to the appendix. Caching is a moderately useful optimization, but not central to Logitext.
>
> > Some of the proposed baselines appear weak and raise concerns about high-quality evaluations.
>
> Please see Issue S4 for details.
>
> > How does the annotation cost compare to using better prompting strategies?
>
> We direct the reviewer to Issue S1, which discusses annotation overhead. In brief, annotation is a one-time cost before deployment similar to extensive prompt optimization techniques, but yields more transparent and controllable reasoning. We are also pursuing automated annotation methods to further reduce this cost.
>
> Overall though, we believe Logitext-style annotation can be a useful tool in the prompt-engineering toolbox. We view traditional prompt-rewriting/refinement strategies as complementary to the logical annotation of Logitext.
>
> > How sensitive is performance to annotation quality and completeness?
>
> We refer the reviewer to Issue S2, which analyzes this question empirically and conceptually. In summary, Logitext performance is surprisingly robust across a range of annotation granularities, and when a decomposition proves unhelpful, it can be easily omitted without compromising the overall system.
>
> However, this question is indeed a critical one and requires a lot more research.
>
> > What happens when logical structures in the document are incomplete or vague?
>
> We direct the reviewer to Issue S5, which discusses this limitation explicitly. Logitext was designed to handle such cases gracefully: authors may formalize only what is clear, and the remainder of the document continues to be processed as natural language.

---

### Official Review · Reviewer_qLbD · 2025-10-31

**Soundness:** 3
**Presentation:** 2
**Contribution:** 3
**Rating:** 6
**Confidence:** 4

**Summary:**

This paper introduces LogiText, a neurosymbolic language which supports partial formalization, and a novel SMT (Satisfiability Modulo Theory) solving framework. This approach aims to bridge the "compositional" and, most notably, the "combinatorial" reasoning gaps that persist in LLMs by coupling an SMT's Boolean search with an iterative LLM-driven "generate-validate-refine" loop.

**Strengths:**

1. This paper proposes a neurosymbolic language, LogiText. LogiText does not require converting the entire document into strict logical formulas; instead, it allows for explicitly annotating only the key logical structures (e.g., Boolean relations) while retaining ambiguous textual clauses as natural language. This design bridges the gap between traditional symbolic solvers (which require fully formalizable domains) and real-world complex documents (which are essentially a mix of text and logic), greatly enhancing the practical value of neurosymbolic methods in domains like legal analysis and content moderation (CMOD).

2. The authors propose a neurosymbolic framework for reasoning with semi-structured language that positions the LLM as an SMT theory solver. Specifically, the SMT symbolic algorithm is responsible for efficient Boolean structure search, and the LLM-driven NLSolver then generates assignments that satisfy logical and semantic constraints by iteratively calling LLM sampling, validation, and refinement operations.

3.  Experimental results demonstrate that on the text instance generation (TIG) and text Coverage Generation (TCG) tasks, the performance of the LogiText-based formalization and the LLM-driven SMT neurosymbolic solving algorithm significantly outperforms that of end-to-end LLMs.

**Weaknesses:**

1. The framework's reliance on precise clause-level annotation and evaluation is a critical vulnerability. LogiText relies on human experts to manually annotate natural language documents. This not only incurs high labor costs but also limits the method's scalability and application scope. Furthermore, the framework is fragile to "clause-level" errors. It still relies on the precise evaluation of each clause, and a failure in evaluating one clause can cause the entire logical chain to collapse. As results on LegalBench (Figure 8) show, a textual judgment error by the LLM on any single clause can lead to reasoning failure. In contrast, the "holistic reasoning" of end-to-end LLMs sometimes exhibits stronger reasoning capabilities. Especially in real, complex scenarios, the partitioning of clauses and the formalization of their logical relationships remain a significant challenge.

2. In the NLSolver algorithm, although the authors introduce a caching mechanism to reduce the number of LLM calls, the cost of LLM calls in the "generate-validate-refine" framework (an iterative refinement process) is still a non-negligible issue. Admittedly, we believe that proposing a low-cost and efficient solving method remains a challenge in this type of search-based neurosymbolic framework. To better assess the practical viability of this framework, we ask the authors to report the average (and maximum) number of LLM calls required by NLSolver to solve each task in the experimental evaluation.

3. Section 3 is hard to follow, partly because the complex notation. Please improve the presentation quality of this part.

(Formatting Issue) Line 412 is incomplete.

**Questions:**

1. The paper states that the benchmarks consist of 15 tasks with "10+ instances each". This seems like a very small scale for evaluation. Could the authors elaborate on the size of the test sets? How can you be confident in the generalizability of the results?

2. The paper mentions using the set of unsatisfied constraints to guide the refinement (Algorithm 1b, line 24). This is a key mechanism. Could the authors provide a concrete example of this refinement prompt? How is the LLM instructed to 'fix' the previously generated text based on which specific natural language constraints failed?

---

> ### Author Response · Authors · 2025-11-20
> **Response to Reviewer qLbD**
>
> We thank the reviewer for the detailed and constructive feedback. Several of the points raised overlap with the cross-cutting issues summarized in our Overall Response Summary, to which we respectfully refer where appropriate. Below we address each concern in turn.
>
> > The method requires labor-intensive clause-level annotation, and such decomposition makes the system fragile to clause-level errors.
>
> These points are discussed in Issues S1 (annotation overhead) and S2 (clause-level robustness) of our overall summary.
>
> In brief, Logitext is intended for deployment-stage prompt engineering rather than casual prompting, where annotation is a manageable part of an existing design process.
>
> Regarding clause-level robustness, our empirical observations show that annotation structures for common documents are favorable for compositional reasoning: subclauses are simpler and typically more reliable, and correlated or “shadowed” clauses mitigate the effect of isolated inconsistencies.
>
> > Even with caching, the "generate-validate-refine" process may be costly. Please report the average and maximum number of LLM calls per task.
>
> In the revised paper, we now include a boxplot summarizing LLM-call statistics (median, maximum, and standard deviation) for the coverage-generation tasks. These results are reported in Appendix A.6.
>
> > Section 3 is hard to follow due to dense notation.
>
> As also discussed in Issue S3, Section 3 has been substantially revised to improve readability. The key changes include:
> - Section 3.1 and 3.2: Simplified descriptions of Logitext constructs with clearer explanations of variable declarations, textual let bindings, and logical constraint blocks. Removed complex subscript notation (e.g., $c_{jh}$, $k_{ih}$) in favor of simpler variable names.
> - Section 3.3: Completely rewrote the solver algorithm presentation to emphasize the intuitive "propose-verify-refine" pattern rather than formal SMT theory. The main text now presents simplified core algorithms that focus on the essential logic, while implementation details (caching mechanisms, history tracking, partial matching) have been moved to the appendix. We introduced explicit function names (LLMPropose and LLMVerify) and made the collaboration between Z3 and the LLM-based NLSolver more transparent.
>
> > Formatting issue: Line 412 is incomplete.
>
> We appreciate the reviewer catching this. The truncation was caused by a LaTeX compilation error and has been corrected in the revised draft.
>
> > The benchmark scale appears small to support generalizable conclusions. Please elaborate on the test-set size.
>
> We refer the reviewer to Issue S4 in the overall summary. In brief we have increased the number of samples by ~10x. Early results indicate that the Logitext performs in a qualitatively similar way as detailed in Appendix A.3.
>
> > Provide a concrete example of how the refinement step prompts the model to “fix” generated text.
>
> We appreciate this suggestion. The revised paper now includes a full refinement prompt and usage examples from the NLSolver in Appendix A.7 (A.7.2 in particular provides a concrete example).

---

### Author Response · Authors · 2025-11-20
**Responses to issues of concern to multiple reviewers**

We thank the reviewers for their comments. We summarize our responses to the main issues here and provide further details to individual reviewers.

## Issue S1: Overhead of hand annotation
The reviewers point out that the additional labor of hand-annotation of documents may make Logitext burdensome, impractical or not scalable.

It is true that for consumer usage (e.g., in ChatGPT), annotating prompts is impractical. In many production scenarios however, prompts are part of deployed programs and undergo extensive "prompt engineering" to maximize accuracy before deployment. We believe that Logitext-style annotations may be a natural tool in this phase. In our experience, adding annotations to even multi-page prompts (e.g., those in the CMOD dataset) may take an hour or so, and can address opportunities that simple textual rephrasing cannot.

We will clarify this focus in our paper.

More broadly, we have ongoing work in auto-annotation. Here we view Logitext as a "target language" where the use of logical solvers as tools for a production prompt can be made explicit and communicated to humans before deployment. So defining a concrete syntax and semantics for the Logitext language, as in this paper, is still valuable even if annotation is automated.


## Issue S2: Fragility to clause-level errors
The reviewers point out that although Logitext may remove errors due to mis-evaluation of logical operations **between** clauses, it introduces new sources of errors via errors in LLM evaluation of **the clauses themselves**.

This is a critical point: we agree it is not clear, **a priori**, that replacing a single call to an LLM with N separate calls that are combined deterministically will yield higher accuracy. E.g., consider a clause C0 which is true 100% of the time by ground truth. Say a single "holistic" LLM call returns true 90% of the time. Say the clause is a conjunct of two subclauses C1 and C2, each of which the LLM separately and independently evaluates to true 90% of the time. Then we would expect Logitext's accuracy to be (0.9)^2 = 81%. Given that real documents may contains tens of relevant clauses, accuracy may plunge.

It is our **empirical** observation that for many natural documents, commonly annotated structures have characteristics favorable for compositional reasoning. To get an idea why, we now include an analysis in Appendix A.4 of the "disruptive behavior" policy of Figure 2. To summarize, for "successful" document/input combinations (the majority in our dataset), we observe:
1. Most subclauses (perhaps because they are simpler) have much higher accuracy rates on most inputs than the parent clause.
2. Subclause misclassifications are not independent: many (successfully classified) inputs tend to have most sub-clauses correctly classified. Many inputs are "obvious".
3. Misclassified clauses may be "shadowed" by others so their errors don't count. In the above example, if clause C0 were **false** 100% of the time, then even if C1 or C2 were mistakenly true, it would suffice if the other were false. The more subclauses, the greater the shadowing effect.
4. Finally, Logitext is complementary to prompt rewriting. Subclauses that are exposed as consistently wrong can be re-written before deployment, a targeted version of traditional prompt engineering. In this submission, we have not done so.

For all these reasons, noise in evaluating subclauses (which we agree is prevalent) is not necessarily catastrophic for Logitext.

## Issue S3: Section 3 is hard to understand
The reviewers especially called out the density of symbols/notation, the fact that Algorithms Check() and NLSolver() were presented in the main text but not explained in full detail, and that definitions for various prompts used in the algorithms were missing.

We have completely rewritten Section 3 to:
1. Reduce the number and complexity of symbols in the text,
2. Move the full definitions of Algorithms Check() and NLSolver() to the appendix and focus in far more detail on their key components in the main text, and
3. Share the full prompts used in the appendix, along with a brief description of them in the main paper.

---

> ### Author Response · Authors · 2025-11-20
> **Responses to issues of concern to multiple reviewers - continued**
>
> ## Issue S4: Evaluation seems underpowered
> Reviewers pointed out that "10+ instances per task" seemed inadequate, wondered if baseline prompting strategy was strong enough.
>
> In response:
> 1. We have increased the number of samples in all tasks to 100 each and provided 3 tables in Appendix A.3 with the number of instances for all 15 tasks, and performance on Logitext on the expanded dataset. In summary, the overall conclusions don't change: Logitext usually provides noticeable boost in compositional reasoning, but "holistic" reasoning occasionally wins because of the noisy-clause-evaluation problem.
>
> 2. We believe our combination of few-shot examples and result justification/chain of thought conforms with best practices for prompting. We share all prompts used in the appendix. We are in the process of using an automated prompt optimization system (DSPy) to improve our baseline, but results are inconclusive as yet.
>
> Regarding prompt optimization, it's worth noting that although we do agree that  better prompting engineering can help close the **compositional** reasoning gap further, it seems unlikely to help with the **combinatorial** gap. A big message of our paper is that although SOTA models do indeed seem to be closing the compositional reasoning gap, there is much further to go with combinatorial reasoning, certainly well beyond what known prompt-engineering can buy us.
>
>
> ## Issue S5: Handling hard-to-formalize concepts and implicit premises
> Reviewers pointed out that many documents contain concepts that are hard to formalize, thus limiting the applicability of Logitext.
>
> We agree with this, and in fact Logitext (as an annotation language within rich documents) was conceived with the full complexity of real-world documents in mind. To understand the Logitext approach, recall that traditional Natural Language Processing researchers have made the case that **compositional logical formulas** can (in principle!) represent the full semantic complexity of language (e.g. via Combinatory Categorial Grammars). This approach failed to scale because the task of producing these parses was never successfully fully automated, precisely when encountering "hard to formalize" concepts such as our reviewers mention. Logitext can be viewed as a practical relaxation of this approach, enabled by the advent of LLMs: we allow you to formalize as much as you can, but leave the rest of the document unformalized. We then provide an algorithm for interpreting any such document.
>
> To the extent that the meaning of a sub-clause depends on text in the rest of the document ("implicit premises"), we note that Logitext's approach of interpreting sub-clause text in the context of the full document is designed precisely to achieve this. We would like to emphasize that we consider the in-context evaluation of predicates to be a strength of Logitext. The context (i.e. the text surrounding the "let" clauses) often includes implicit premises, and Logitext's clause evaluation takes this context into account even if it is outside the "let" binding.
>
> **This example added after discussion with reviewer AF7g** For instance, consider a policy document titled "Privacy Policy for Mobile Device Data". The body of the document may contain a clause requiring that "the stored data should not include sensitive information". Although the phrase "sensitive information" is very broad, the "implicit premise" from the title above narrows it to privacy sensitivity. Logitext is naturally able to handle such context because NLTCs include the original document d as part of their definition and the clause in the NLTC is evaluated in the context of the corresponding document.
>
> We admit that it is entirely possible that in many documents of interest, important but problematic (i.e., hard to infer reliably with LLMs) sections may not easily be formalizable with Logitext notation. In this paper, we only take a first step in this direction to show that for many types of documents of interest it **is** indeed possible to formalize partially but profitably even using Logitext's simple propositional logic based formulation. How to expand the set of applicable documents is future research.

---

### Author Response · Authors · 2025-12-03
**Summary Comment After Discussion Session**

We thank the reviewers for their feedback, which has already helped us improve our work significantly.

We have shared detailed responses below and attached a substantially revised version of the paper. We have also attached a "diff" version of the new draft in supplementary material.

Somewhat unusually, we have both strong opposition and strong support for the paper. We believe the strong opposition is in large part due to very specific misconceptions:

* Reviewer EsVD rated our Contribution "poor" (score 1), whereas the others rated it 3, 3, 4. It seems EsVD's concern is that Logitext's manual annotation of prompts is a not scalable and requires a contrived workflow. This seems to be the dealbreaker, and the primary reason for the strong reject.

  We agree that *for consumer prompting* (e.g., chat systems), annotation is impractical. However, as we point out below, *in production deployments* (e.g., when deploying the example content moderation policy in our paper), prompts are embedded in code and reused millions of times. Prompt engineering, which requires time-consuming prompt editing and testing, is common in this case, and annotation a la Logitext is quite reasonable.

* Reviewer AF7g rates our Presentation poor (other reviewers share complaints about our overly complex presentation), and rates the Soundness as fair (score 2); we thank them for the 3 on Contribution! It's unclear if one or both of these was the dealbreaker, and what merited the strong reject.

  Re Presentation, we have incorporated feedback on improving presentation and believe our updated draft is much improved as a result.

  If we understand correctly, AF7g's concern on soundness comes from the concern over whether Logitext's need for *explicit* annotations of clauses means that *implicit* premises cannot be satisfied. As we explain in our response to AF7g's response, Logitext does indeed allow implicit context/premises to be incorporated in a powerful and natural way. It is unclear what other forms of implicit context we are missing and would appreciate any specific examples.

We are hopeful that the reviewers will reconsider the strong rejects given the above clarifications. We believe Logitext does not violate the apparent dealbreakers. If strong objections remain, we request more details (e.g. examples) so we can address these issues in our work for the future.

In any case, we again thank **all** the reviewers for their thoughtful and detailed feedback, all of which we have tried to address or incorporate.

---

### Meta-Review · Area_Chair_u4et · 2026-01-07

**Summary:**

This paper proposes Logitext, a neuro-symbolic language that aims to bridge the gap between natural language and formal language. Logitext allows the logical structure of text to be explicitly represented in formal notation and is internally modeled using a novel form of natural language constraints. Experiments on 15 tasks demonstrate the effectiveness of the proposed method. Overall, this is an innovative contribution to logical reasoning, and the experimental results show that the LLM-driven SMT-based neuro-symbolic solving algorithm significantly outperforms end-to-end LLM baselines.

During the rebuttal, reviewers raised concerns regarding the cost of human annotation, the convergence guarantees of the NLSolver, and several presentation issues, including notation clarity and figure readability.

**Reviewer Concerns:**

The authors respond to most concerns regarding the cost of human annotation, clarify several technical details, and revise presentation issues, including notations and figures. However, concerns remain about the Logitext's lack of a principled and compelling bridge between natural and formal language, the scalability of manual annotation, and the need for further improvements in presentation.

**Reviewer Scores:**

The clarifications provided in the rebuttal address several reviewer concerns and may warrant a slight increase. Overall, given the remaining issues, I consider this paper to be borderline.

---

### Decision · Program_Chairs · 2026-01-26

Reject